# Electrodeposited Zinc Coatings for Biomedical Application: Morphology, Corrosion and Biological Behaviour

**DOI:** 10.3390/ma16175985

**Published:** 2023-08-31

**Authors:** Purificación Tamurejo-Alonso, María Luisa González-Martín, Miguel Ángel Pacha-Olivenza

**Affiliations:** 1Department of Biomedical Science, Faculty of Medicine and Health Sciences, University of Extremadura, 06006 Badajoz, Spain; ptamurejo@unex.es; 2University Institute of Extremadura Sanity Research (INUBE), 06006 Badajoz, Spain; mlglez@unex.es; 3Department of Applied Physics, Faculty of Science, University of Extremadura, 06006 Badajoz, Spain; 4Networking Research Center on Bioengineering, Biomaterials and Nanomedicine (CIBER-BBN), 06006 Badajoz, Spain

**Keywords:** implant, zinc, coating, electrodeposition, morphology, corrosion, antimicrobial, osseointegration

## Abstract

The improvement of biodegradable metals is currently an active and promising research area for their capabilities in implant manufacturing. However, controlling their degradation rate once their surface is in contact with the physiological media is a challenge. Surface treatments are in the way of addressing the improvement of this control. Zinc is a biocompatible metal present in the human body as well as a metal widely used in coatings to prevent corrosion, due to its well-known metal protective action. These two outstanding characteristics make zinc coating worthy of consideration to improve the degradation behaviour of implants. Electrodeposition is one of the most practical and common technologies to create protective zinc coatings on metals. This article aims to review the effect of the different parameters involved in the electrochemical process on the topography and corrosion characteristics of the zinc coating. However, certainly, it also provides an actual and comprehensive description of the state-of-the-art of the use of electrodeposited zinc for biomedical applications, focusing on their capacity to protect against bacterial colonization and to allow cell adhesion and proliferation.

## 1. Introduction

Zinc is a ductile and malleable metal of atomic number 30 found in the earth’s crust at concentrations of approximately 50 ppm [1]. It is chemically active and reacts with non-oxidizing acids, releasing hydrogen to form zinc ions (Zn^2+^). Dry air does not attack it, but humid air causes it to oxidize, generating a hard film on its zinc carbonate surface, which protects it from further corrosion. For this reason, it is highly valued in the industry as a coating for other metals to protect them from corrosion. Its effectiveness is due to its barrier effect. The protective Zn layer prevents direct contact of the material in the lower layer with the surrounding medium species [2,3] by causing a cathodic sacrificial reaction in the Zinc which prevents the coated material from corroding [2,4,5]. Its resistance to corrosion depends largely on the morphology, topography and texture it shows in the coating [6]. At room temperature, it is a metal that crystallizes into a compact hexagonal structure [7] with a maximum packing factor. The growth plane can be basal, pyramidal or prismatic. This results in different crystallographic orientations that will determine whether the crystalline surface is more or less exposed to the surrounding medium [8] and will result in a characteristic morphology. In addition, a reduction in roughness can lead to deposits with a fine grain size. Consequently, promoting the reduction in the contact area between the zinc deposit and the corrosive environment [3,6] will favor anticorrosive behaviour. In very aggressive environments or at high temperatures, the protection provided by zinc is not sufficient [9]. For this reason, zinc is alloyed with more noble ferrous metals such as cobalt, nickel and iron [5,10], which provide better corrosion resistance than pure zinc. In particular, the ZnNi alloy has attracted the attention of the automotive [11], electronics [12] and even the marine industry [13] due to its ease of workability, low cost and good corrosion-resistance properties, all consequences of the sacrificial anodic behaviour for nickel content between 12–15% [14].

An interesting application that is attracting the attention of researchers is the development of coatings with zinc-based materials for implantable medical devices [15]. Their anticorrosive properties prevent the release of metal ions around the implantation site and thus prevents inflammatory reactions in the body. Biodegradable implantable devices are designed to go unnoticed while the damaged tissue heals without additional surgery [16]. Some authors have shown interest in zinc in its oxide form (ZnO) [17,18,19], which is an n-type semiconductor with a bandgap energy of 3.37 eV [20], similar to that of TiO_2_ (3.2 eV) [21], so it can be excited by light in the ultraviolet (UV) spectrum. This induces photocatalytic surface reactions such as the photogeneration of reactive oxygen species (ROS).

The total zinc content in the human body ranges from 2–4 g, however, its concentration in plasma is as low as 12–16 μM, which gives it the name of a trace element [22]. There is no particular site within the human body that acts as a reservoir for Zn, which makes its daily intake necessary. Zinc is involved in many of the processes that regulate human physiology. It is involved in gene expression, cell division, growth, wound healing and cellular immunity [23]. It promotes the processes of osteoblastic bone formation and mineralization [24]. Zinc deficiency in the human body has been associated with skeletal abnormalities and delayed bone development [25,26]. It is also used in the pharmaceutical industry because of its important role in human health.

A decisive factor for the success of an implant, both in permanent and biodegradable materials, is bacterial contamination. It can originate at the time of surgery due to contamination of the skin flora of the patient, the healthcare staff or even the operating theatre environment [27]. However, they can also originate from a distant source of infection [28]. The initial adhesion of bacteria to the surface is a reversible process, but when biochemical interactions between the surface of the material and the bacteria start to act, the process becomes irreversible. In these cases, bacteria start to develop their protective mechanism against antibiotic action, biofilms, leading to the removal of the implant [29]. On the one hand, the Zn^2+^ ions released by the coating when partially dissolved in an aqueous medium have a bacteriostatic effect. The peptidoglycan layer-forming part of the bacterial wall is negatively charged and the Zn^2+^ ions are trapped in it [30]. This results in the detection of bacterial metabolism, but without causing death. On the other hand, under the action of UV light, the interaction of ROS with cells triggers compensatory mechanisms. When there is an imbalance between the ROS generated and their reduction, oxidative stress occurs [30,31,32,33,34,35,36,37], which causes a structural and functional change in the micro-organism, accelerating its ageing and promoting cell death. The biocidal activity of these ROS is directly related to the generation of H_2_O_2_. This is the only product that can penetrate cytoplasmic membranes [38,39,40,41], causing lipid peroxidation. Once the membrane is damaged, the rest of the ROS products penetrate the membrane, causing a further biocidal effect [42,43,44,45,46,47,48,49,50,51]. This is due to the existence of a greater or lesser number of surface defects on the crystalline ZnO particles.

One of the most common procedures to achieve complete and uniform coatings in flexible and economical way is electrodeposition. The Zn electrodeposition process consists of generating a metallic coating on a base material, even on pure Zn [52], thanks to the electrochemical reduction in the electroactive species dissolved in an electrolyte. This process consists of several steps [53,54,55]. Hydrated ions and complex ions migrate from the bulk of the electrolyte solution to the interface. At the interface, the electroactive species undergo a chemical conversion reaction such as ligand detachment or dehydration, followed by a charge transfer process. The reacting particles gain electrons to become atoms adsorbed on the electrode surface. These two processes are decisive in determining the morphology, as well as the anti-corrosion properties of the resulting coating, just because the atoms generated in the chemical conversion reaction come together to form crystalline nuclei that will gradually grow, or they move across the electrode surface, reach a specific position and enter the lattice to continue growing in the original lattice [56,57]. As the electrodeposition process proceeds, some clusters can be stable or unstable. The unstable ones will eventually disappear, while the stable ones will grow and form the film [58]. By controlling the various parameters involved in the process, a firm and compact electrodeposit is achieved at the cathode. There are several parameters that allow the control of electrodeposition and influence the final finish of the deposit, and therefore, on the properties of zinc coatings.

Within this scope, the aim of this literature review is to provide the reader with an update on the influence of the modification of the different parameters that can be adjusted over the course of the electrodeposition process, such as the current density, temperature or additives, and the effect that the characteristics of the electrodeposited layer have on different kind of materials. Moreover, properties such as antibacterial activity and osseointegration of Zn electrocoatings are revised in materials directly designed for the fabrication of implants.

## 2. Surface Morphology of the Zn Coating

Several experimental variables in the electrodeposition process are decisive in the morphology of the electrodeposited coating, since they can increase the number of active sites for nucleation or favour the absorption of the electroactive species. These include current density and application method, temperature, hydrogen gas evolution, electroactive ions and the use of additives. Table 1 shows a summary of the literature on the influence of the electrochemical parameters of Zn deposition on the morphology.

### 2.1. Current Density

Current density is a parameter that, during the Zn electrodeposition process, allows for controlling the rate at which the deposition occurs with a significant influence on the nucleation processes and, therefore, on the resulting morphology [59,60]. The electrodeposition technique can be carried out using a direct current (DC), where the only variable to be controlled is the current density [61], or it can be carried out using a pulsed current (PC), with a higher complexity of the process, as it involves a larger number of experimental variables: on-time (T_on_), off-time (T_off_), duty cycle and pulse peak current density (J_p_). Focusing on DC electrodeposition, Freitas et al. [62] analysed the influence of current density on the crystallization of Zn deposits in steel sheets, and obtained that, for high current densities of 50 mA/cm^2^, the growth rate of the nuclei radius in a direction parallel to the electrode surface was higher than in other directions. With this orientation, the overlapping of the nuclei was favoured and therefore a decrease in the presence of microporosities took place. However, for low densities of 10 mA/cm^2^, the growth rate of the nuclei radius in a direction perpendicular to the electrode surface was higher than in other directions, and the nuclei radio did not overlap, resulting in high microporosity. N. Alias et al. [63] also studied the influence of different current densities on Zn electrodeposits on a copper surface from a zinc sulphate solution. These authors found a thin Zn deposit with little nucleation at a low current density, 10 mA/cm^2^. However, the morphology of the deposit changed to a very flat platelet-like hexagonal crystal structure layer-by-layer at a current density of 20 mA/cm^2^, leading to a dense nucleation with irregular growth of metallic Zn at 40 mA/cm^2^, and even allowing the development of three-dimensional flake-like structures at 100 mA/cm^2^. Changes in the topography of the Zn electrodeposited on mild steel because of the modification of the current density were also corroborated by Fazazi et al. [64]. At low current densities, around 16 mA/cm^2^, the morphology of the deposit was smooth, platelet-like and with compact hexagonal crystal grains. However, at current densities higher than 40 mA/cm^2^, deposits evolved to a heterogeneous, flower-shaped three-dimensional structure with a rough topography.

The influence of current density on the characteristics of the coating is also evident when nanoparticles are included in the surface layer together with the zinc compounds. Interest in this procedure is because nanoparticles in the electrodeposited coating improve corrosion resistance, microhardness, wear and even self-lubrication and can modify the morphology of the resulting coverage depending on the applied current density [65,66,67,68,69,70,71]. Cabral-Miramontes et al. [72] probed that the inclusion of Zn/TiO_2_ and Zn/ZnO nanoparticles changes the morphology of the electrodeposited zinc in respect to the exclusively zinc coating, in a way that is dependent on the applied current density. In the case of a current density of 50 mA/cm^2^, the morphology of a pure Zn coating has a hexagonal crystal structure, but the composite coatings included uncoated areas with heterogeneous and non-uniform crystals. However, when the current density was higher than 100 mA/cm^2^, both coatings showed a compact and homogeneous hexagonal crystal morphology, having a smaller grain size in the coatings with nanoparticles than in the case of pure Zn, probably because the addition of nanoparticles caused a faster formation of nucleation sites, thus reducing the grain size. For both current densities, thickness was higher for the pure Zn coatings. In this line, Camargo et al. [73] prepared Zn coatings with TiO_2_ nanoparticles at a concentration of 15 g/L on a steel substrate with DC densities of 20 mA/cm^2^ and 200 mA/cm^2^. Moreover, in this research, the higher current density provides a more uniform and compact morphology, with TiO_2_ agglomerates adsorbed on the surface of the deposit, than when current density was 20 mA/cm^2^. In this last condition, the porous morphology of the coating allows TiO_2_ particles not only on the surface but also inside the pores. Further research by these authors [74] combined ultrasonic power density combinations with a variation in current density in the electrodeposition process of Zn/TiO_2_ coverages. When the current density was low, 20 mA/cm^2^, without ultrasonic irradiation, the crystal morphology was irregular and coarse-grained. However, when combined with an ultrasonic current density of 28 mW/cm^3^ or 53 mW/cm^3^, it showed a levelling effect on the morphology, with a roughness smoothing of Ra 2.5 mm and 1.5 mm, respectively. At higher values of current density, the levelling effect gradually decreased up to 200 mA/cm^2^, for which the influence of ultrasonic irradiation was negligible.

When the pulsed current (PC) method is used, together with the pulse peak current density (J_p_), the on-time (T_on_), off-time (T_off_) and duty cycle can affect the microstructure and properties of the resulting coating. Kim et al. [75] and Sun et al. [76] investigated the influence of T_on_ and T_off_ on Ni electrodeposition by varying the pulse frequency (f = 1/T_on_ + T_off_) and keeping J_p_ constant. Nevertheless, El-Sherik et al. [77] stated that since the pulse frequency is a magnitude obtained from the original parameters T_on_ and T_off_, that procedure will never be able to reflect the separate effects of T_on_ and T_off_. Puippe et al. [78] examined different values of J_p_ at a constant T_on_ and a constant average current density. However, this method had the disadvantage that T_off_ could not be kept constant, and the coverage results could be attributed to the increase in J_p_ or the increase in T_off_. On the other hand, Youseff et al. [79] investigated the effect of a single parameter (T_on_, T_off_ or J_p_), keeping the other two constant in the influence of the grain size and morphology of nanocrystalline Zn coatings. When they kept T_on_ = 5 ms and T_off_ = 9 ms constant and varied J_p_ in the range of 400 mA/cm^2^ to 1000 mA/cm^2^, they found that low values of J_p_ generated large grain sizes in the form of plates. However, when they increased the J_p_ value from 800 mA/cm^2^, a drastic reduction in the spherical grain size took place on the average order of 50 nm. Moreover, Kartal et al. [80] found that increasing J_p_ and keeping T_on_ and T_off_ (both 5 ms) constant resulted in a decrease in crystal size on the microstructure of Zn deposits. This result was confirmed by other authors [78,81]. However, when studying the influence of T_on_ and T_off_, keeping all other parameters constant, it is not always possible to predict the characteristics of the deposit in each system, as each may react differently during the electrocrystallisation process [78].

Moreover, it can be expected that the results of using a given current density is modulated if the electrodeposition process uses a DC or PC. Deo et al. [82] fabricated electrodeposited ZnMn alloy coatings on steel substrate with DC and PC densities of 60 mA/cm^2^. When the applied current density was DC, they observed a thin plate morphology with a cauliflower-like appearance, as described by other authors [83]. However, when the current density was PC, the rough cauliflower morphology was transformed into a smooth globular structure due to the replenishment of ions near the cathode surface during T_off_. They also observed that a higher working frequency led to higher T_off_, and therefore more interruptions in the deposition process. This resulted in a greater preference for nucleation over crystal growth, leading to a refinement of the grain size, as Claudel et al. also observed [84]. In the same line, Kancharla et al. [85] developed zinc electrodeposition on steel using DC and PC at different current densities (10 mA/cm^2^, 30 mA/cm^2^ and 60 mA/cm^2^), but keeping T_on_ and T_off_ constant (T_on_ = 60 ms and T_off_ = 240 ms). A uniform distribution of Zn crystals was observed in all conditions. The average grain size of the coatings deposited with DC at 10 mA/cm^2^, 30 mA/cm^2^ and 60 mA/cm^2^ was 5.95 ± 0.65 µm, 5.80 ± 0.44 µm and 5.73 ± 0.041 µm, respectively, but the average grain size in the coatings prepared by PC were slightly finer, with a size of 4.20 ± 0.30 µm, 4.01 ± 0.33 µm and 3.89 ± 0.45 µm, respectively. Ultimately, a higher J_p_ resulted in a higher nucleation rate on the cathode surface, leading to inhibition of crystal growth [59,86]. The authors also determined the average roughness values (Ra) using DC at 10 mA/cm^2^, 30 mA/cm^2^ and 60 mA/cm^2^ and obtained values of 111 nm, 110 nm and 112 nm, respectively. Similarly, the average values obtained using PC were 112 nm, 113 nm and 115 nm, respectively. When comparing the Ra values, the deposits by PC showed slightly higher values than by DC. Similar results were obtained for the topographic parameters of mean maximum height (Rz). This indicated a higher roughness obtained in the PC coatings than in the DC coatings, despite the finer and more compact morphology of the former. The authors also studied the symmetry of the coating through the R_sk_ parameter. The R_sk_ values with DC at 10 mA/cm^2^, 30 mA/cm^2^ and 60 mA/cm^2^ were 0.157, 0.014 and 0.179, respectively. Similarly, the R_sk_ values with PC were 0.086, 0.007 and 0.119, respectively. This denotes that the PC-coated samples present a more symmetrical morphology than the DC-coated samples.

### 2.2. Temperature

The temperature at which the electrodeposition process takes place will also significantly influence the morphology of the resulting Zn coating. When the process takes place at elevated temperatures, solubility and electrical conductivity improve and anodic passivation is reduced, but it also accelerates solution evaporation and corrosion processes, slowing down diffusion kinetics, often resulting in deposits prone to embrittlement [87]. Zhang et al. [88] electrodeposited pure Zn on Al sheet using DC. They analysed the influence of temperature, between 30 °C and 45 °C, on the final morphology of the coating, formed by hexagonal plates. With increasing temperature, the size of the deposit formed progressively decreased for all the different values of current densities tested (300 A/m^2^, 400 A/m^2^, 500 A/m^2^ and 600 A/m^2^). Qiao et al. [89] showed that the temperature at which the coating was produced, in a range between 10 °C and 70 °C, had a large effect on the morphology of ZnNi electrodeposited on a carbon steel sheet at a current density of 15 A/cm^2^ using DC. Temperature increases up to 40 °C caused the coating to become increasingly uniform and compact, and the crystalline morphologies changed from a loose cauliflower to a fine-grained pyramidal structure. Above 40 °C, the compactness of the coating started to degrade, with surface voids and cracked regions emerging, and with crystalline morphology changing to a quadrangular configuration, probably due to a hydrogen release [90] that evolves with the increasing temperature and reaches a maximum at 70 °C.

The morphology of ZnO coatings is also conditioned by the electrodeposition temperature. Mentar et al. [91] electrodeposited ZnO nanostructures on glass substrates at temperatures between 30 °C and 70 °C and with a fixed applied potential of −1.3 V with respect to the saturated calomel electrode. Electron microscopy results showed different morphologies with temperature. At 30 °C, thin nanolayer structures were identified. When the temperature increased from 40 °C to 70 °C, some lamellar structures became quite large. At 50 °C, the morphology exhibited interlocked nanostructure, widely described in the literature [92]. Moreover, increasing the temperature of the electrolytic bath caused the current density to increase from −1.21 mA/cm^2^ to −3.38 mA/cm^2^. Similar behaviour was observed by Goux et al. [93] in the range of 34 °C and 89 °C, with a constant potential of −0.75 V with respect to the normal hydrogen electrode. At 34 °C, the ZnO coating was formed by a set of grains without defined crystallography and with porosity typical of a slow nucleation. Between 40 °C and 80 °C, the ZnO layers became dense, with very low roughness and complete coverage of the substrate, typical of optimal crystallinity. Surprisingly, at 89 °C a change in the film morphology occurred. Small, elongated crystals with high roughness were observed. The authors concluded that crystallisation occurred at 34 °C, but the nucleation was not instantaneous but a delayed process that appears after a period of induction. Otani et al. [94] tested temperatures between 25 °C and 70 °C and a fixed potential of −0.8 V with respect to the silver chloride reference electrode. At 25 °C, the presence of ZnO with low crystallinity was practically undetectable. As the temperature increased, the crystallinity improved markedly. Hexagonal structures with a preferential orientation parallel to the substrate started to appear. Saidi et al. [95] also corroborated these temperature dependencies. They stated that low electrodeposition temperatures of ZnO required the need to form intermediate precursors for initial nucleation and subsequent crystal growth to take place. However, higher temperatures resulted in direct nucleation without the need for such precursors. This change was attributed to the higher stability exhibited by ZnO vs. Zn(OH)_2_ at elevated temperatures.

### 2.3. Additives

#### 2.3.1. Hydrogen Peroxide

The production of hydrogen bubbles because of the dissociation of the water present in electrolytes is a phenomenon that must be considered in the morphology of the resulting coating by electrodeposition. Some hydrogen atoms are adsorbed on the surface of the coating, diffusing into the crystal lattice of the deposit, and causing a high residual stress to develop. This process is favoured by the increase in temperature, generating the appearance of macroscopic point defects such as pits, pores or cracks that compromise the homogeneity of the electrodeposited coating [90]. It is known that the addition of H_2_O_2_ in the preparation of the electrolyte causes the reduction in hydrogen bubbles [96] and then favours the access of new zinc ions to active nuclei sites in the cathode surface. Therefore, it seems logical to think that an optimal H_2_O_2_ concentration will cause the highest density of active nuclei for a given electrodeposition process [87]. Pauporté et al. [97] studied the influence of different concentrations of H_2_O_2_ (2.5 mM to 40 mM) as an oxygen precursor in the electrodeposition of ZnO on a glass slide with a conductive SnO_2_ film. The coating was performed at 70 °C and at a deposition potential of −1.4 V with respect to the HgSO_4_ reference electrode. When the coatings were prepared at concentrations up to 25 mM, they showed well-defined crystals; specifically, at 5 mM, columnar crystalline structures with a hexagonal cross-section and very sharp edges were observed. Gradually increasing the concentration caused a decrease in crystal size and a detrimental effect on edge sharpness. At 40 mM, the appearance changed to a very flat granular surface. Higher H_2_O_2_ concentrations achieved higher deposition rates. Similar morphological changes were detected by Henni et al. [98]. In that case, they obtained ZnO deposits on glass slides with In_2_O_3_ conductive film at different H_2_O_2_ concentrations (2–15 mM). The coating was carried out at 65 °C with a potential of 1.0 V with respect to the calomel electrode. For all H_2_O_2_ concentrations, the morphology of the ZnO obtained was in the form of hexagonal-shaped nanorods. For values of 2 mM H_2_O_2_, the hexagonal nanorods were practically perpendicular to the substrate with a relatively low density. Higher concentrations up to 7 mM resulted in a higher density of nanorods with a smooth surface and diameter around 115 nm to 185 nm. A higher concentration between 10–15 mM resulted in an excess of hydroxide ions on the electrode surface, which caused the zinc ions present in the vicinity of the electrode to be consumed very quickly. This would result in a high rate of electrodeposition. Under these conditions, a multidirectional growth of ZnO nanorods took place. The authors confirmed that higher H_2_O_2_ concentrations led to a higher nucleation density and a higher density of ZnO nanorods.

#### 2.3.2. Other Additives

Incorporating additives into the electrolyte for zinc electrodeposition also influences the morphology and grain size refinement of the coating [96,99,100,101,102]. The specific mechanism for the action of the additives in the zinc electrodeposition process is not fully determined, but a number of facts occur that can help in understanding their behaviour during coating fabrication: (a) most additives present a net positive charge and migrate to the cathode, where they are absorbed; (b) their analytical confirmation is very difficult because their actual presence in the deposit is extremely small; (c) the growth of columnar deposits is eliminated by their presence in the electrolyte; (d) the composition of the electrolyte tends to alter the electrochemistry of the organic additive [103]. Organic additives such as surfactants are used in electrodeposition solutions to improve particle distribution in the coating and to facilitate the deposition of the composite coatings [104,105,106,107]. They influence the absorption process at the cathode surface and can therefore have large effects on the electron transfer kinetics. This includes blocking of active sites and possible interactions between the electroactive species and themselves [108]. For this reason, the additives induce changes in the orientation as well as in the morphology of the deposit. Saber et al. [81] used the organic additives polyacrylamide and thiourea in a PC electrodeposition of zinc on low carbon steel. The process was carried out with different J_p_ ranging from 0.4 A/cm^2^ to 2 A/cm^2^, keeping T_on_ and T_off_ fixed at 0.1 ms and 1 ms, respectively. They determined that, in the presence of these additives, refinement was achieved by increasing the J_p_ at a concentration of polyacrylamide and thiourea between 0.2–1.5 g/L and 0.02–0.5 g/L, respectively. Working with the same additives, Youssef et al. [79] reported the grain refinement of a zinc deposit on low carbon steel by increasing the T_on_ but keeping the T_off_ and J_p_ fixed. In this case, the optimum concentrations of polyacrylamide and thiourea would be 0.7 g/L and 0.05 g/L, respectively.

On the other hand, Gomes et al. [109] also prepared zinc deposits by electrodeposition of PC from acid solutions of zinc sulphate on stainless steel. These authors focused on studying the influence of anionic (sodium dodecylsulphate, SDS), cationic (cetyltrimethylammonium bromide, CTAB) and non-ionic (octylphenolpoly(ethylene glycol ether)n, n = 10, Triton X-100) surfactants on the morphological characteristics of the electrodeposited coatings. The electrodeposition PC parameters were 100 mA/cm^2^, J_p_; 4 ms, T_on_ and 40 ms, T_off_. The authors observed that the presence and nature of the surfactants used promoted coatings with different texture, crystalline shape and grain size (between 40 nm and 20 nm). Deposits prepared in the presence of SDS were very uniform, with plate-like crystals oriented perpendicular to the substrate. Compared to the deposits prepared without additives, they showed relatively smaller grains, in line with results of other authors [110]. The deposits prepared in the presence of CTAB showed a change in morphology with respect to the previous one. They were porous due to needle-shaped grains with a smaller size. This shows a blocking effect of the cationic surfactant, leading to an increase in core renewal rates. In the presence of Triton X-100, the coating was more irregular, with a different morphology from the previous ones, consisting of cauliflower-like agglomerates. The non-ionic surfactant had a strong influence on the Zn electrodeposition process. Trejo et al. [111] reported similar morphology in the presence of organic polyethoxylate, as ethyleneglycol and their polymers (PEG) but depending on their molecular weight.

Polypyrrole (PPy) has been used to achieve a homogeneous dispersion of the nanoparticles in the electrodeposited coating. Maimaiti et al. [112], tested this additive from the in situ oxidative polymerisation of pyrrole (Py) to achieve a homogeneous distribution of hydroxyapatite (HA) and ZnO nanoparticles on the surface of a titanium substrate. Deposits prepared with a Py concentration of 0.01 mol/L showed a coating in the form of nanoneedles because of a too low amount of PPy formed by electrochemical polymerisation. When the deposit was prepared with a concentration of 0.03 mol/L, the coating was composed of spherical nanoparticles with relative uniformity. A higher concentration, around 0.05 mol/L, caused the morphology to change to a network of nanorods with uneven distribution and size. This was possibly because the Py concentration was too high and excess Py molecules would oxidise, leading to polymerisation into PPy macromolecules under the action of an electric field before reaching the region of the electrode where it should react. The authors concluded that for effective PPy production to take place and in order to employ the electrochemical polymerisation method, the Py concentration had to be in an appropriate range.

### 2.4. Electroactive Ions

The concentration of electroactive ions used during the electrodeposition process plays an important role in the final morphology of the coating, namely the grain refinement and the rate of Zn deposition. He et al. [113] fabricated ZnFe alloy coatings on a mild steel sheet using an electrolyte of ZnSO_4_ and FeSO_4_ with different Zn^2+^/(Fe^2+^ + Zn^2+^) ratios of 1/10, 2/10 and 3/10. The electrodeposition was carried out under DC densities of 10 mA/cm^2^ and 20 mA/cm^2^ at a fixed temperature of 50 °C. Irrespective of the current density, the coatings showed different microstructures from rough band to a particulate morphology to a plain structure with the increase or the ion ratios.

The effect of different concentration of electroactive ions was also reflected in the systems studied by M. Guo et al. [114]. They obtained nanorods of ZnO on glass using an aqueous ZnCl_2_ solution with different concentrations of Zn^2+^ at a fixed potential of −1.0 V with respect to the calomel electrode and a temperature of 70 °C. Relatively high ion concentrations of 0.005 mol/dm^3^ led to the appearance of dense deposits without voids. When the concentration decreased to 0.001 mol/dm^3^, arrays of ZnO nanosheets with a relatively higher density and smaller diameter of 21 nm were observed. When the concentration became even smaller, around 0.0005 mol/dm^3^, the diameter of the nanorods also decreased to 18 nm and the density increased. The low degree of supersaturation is believed to be responsible for the small diameter and narrow size distribution of the electrodeposited ZnO nanorod. The authors justified this relationship on the basis that the electrodeposition of ZnO will depend mainly on Zn ion concentrations, and only when the zinc concentration is lower than 0.015 mol/dm^3^ can there be a possibility of the formation of a ZnO nanorod. If the concentration of zinc ions is high, their diffusion will be large, and a large amount of zinc ions will accumulate on the surface of the substrate. This will cause the electrodeposition rate to be higher. Therefore, the average diameter of ZnO nanorods will increase [115].

Izaki et al. [116] reported that from solutions with Zn(NO_3_)_2_ concentration between 30 mM and 100 mM, a well-defined orientation was obtained for ZnO films deposited on glass slides. However, if the concentration was increased, the orientation was lost. In a similar system, Illy et al. [117], obtained coatings composed of rod-shaped particles of ZnO. Concentrations between 60–90 mM of Zn(NO_3_)_2_ generated rods with a perpendicular orientation to the substrate and high density. From 90 mM onwards, the rods lost their perpendicular orientation, which led to the appearance of multidirectionality.
materials-16-05985-t001_Table 1Table 1Summary of the literature on the influence of electrochemical parameters on the morphology.Ref.SurfaceCoatingCurrentTypeCurrent Density(mA/cm^2^)Morphology[62]SteelPure ZnDC10Grain growth in direction perpendicular. High microporosity50Grain growth in parallel direction. Low microporosity[63]CopperPure ZnDC10Little nucleation20Dense nucleation. Platelet-like hexagonal crystal structure40Dense nucleation. Irregular growth100Three-dimensional flake-like structures[64]Mild SteelPure ZnDC16Platelet-like with compact hexagonal crystal grains>40Three-dimensional structure with flower-shaped[72]1018 carbon steelPure Zn DC50Hexagonal crystal structureHeterogeneous and non-uniform crystals>100Hexagonal crystal structure with large grainZn/TiO_2_ andZn/ZnOCompact and homogeneous hexagonal crystal with small grain[73]SteelZn/TiO_2_DC20Porous morphology with TiO_2_ particles on the surface and pores200Uniform and compact morphology with TiO_2_ agglomerates[82]SteelZnMnDC60Cauliflower-like roughPC(J_p_ 60 mA/cm^2^)Smooth globular structure. Higher T_off_ causes better nucleation and worse crystals growth[85]SteelPure ZnDC10Higher current density (DC) causes smaller grain size30PC(T_on_ 60 ms and T_off_ 240 ms)Higher current density (PC) causes smaller grain size. Higher grain refinement with PC versus DC60**Ref.****Surface****Coating****Current****Type****Temperature****(°C)****Morphology**[88]AluminumPure ZnDC30Hexagonal crystal structure progressively decreasing in size with increasing temperature45[89]Carbon SteelZnNiDC10 < T < 40Change of cauliflower morphology to fine-grained pyramidal structure. Uniform and compact40 < T < 70Coating degradation. Surface voids and cracked regions[91]GlassZnODC30Thin nanolayer structures50Interlocked nanostructure70Large lamellar structures[93]GlassZnODC34Grains without crystallography and with porosity40 < T < 80Good crystallography. Low roughness and dense coverage89Small, elongated crystals with high roughness[94]GlassZnODC25Very low crystallinity70Improved crystallinity. Hexagonal structures with preferential orientation parallel**Ref.****Surface****Coating****Current****Type****Additives****Morphology**[97]Glass with SnO_2_ filmZnODCH_2_O_2_5 mMColumnar hexagonal crystal structure. Sharp edges25 mMColumnar hexagonal crystal structure. Smaller grain size40 mMVery flat granular surface[98]Glass with In_2_O_3_ filmZnODCH_2_O_2_
2 mMLow density of perpendicular hexagonal nanorods crystals7 mMHigher density of perpendicular hexagonal nanorods crystals15 mMMultidirectional growth nanorods crystals[81]Low carbon steelPure ZnPC (T_on_ 0.1 ms and T_off_ 1 ms)Polyacrylamide (0.2–1.5 g/L)Grain refinement at these additive concentrations and at higher J_p_Thiourea (0.02–0.5 g/L)[79]Low carbon steelPure ZnPC (T_off_ 9 ms and J_p_ 800 mA/cm^2^)Polyacrylamide(0.7 g/L)Grain refinement at these additive concentrations and higher T_on_Thiourea (0.05 g/L)[109]Stainless steelPure ZnPC(J_p_ 100 mA/cm^2^, T_on_ 4 ms and T_off_ 40 ms and)SDSDeposits uniform with plate-like crystals oriented perpendicularCTABPorous. Needle-shaped grains with a smaller sizeTriton X-100Irregular coating. Cauliflower-like agglomerates[112]TitaniumHA/ZnOPC(pulse potential of 1.0 V/−2.5 V)Py0.01 mol/LNanoneedles0.03 mol/LSpherical nanoparticles0.05 mol/LNanorods with uneven distribution and size**Ref.****Surface****Coating****Current****Type****Electroactive ions****Morphology**[113]Mild steelZnFeDCZn^2+^/(Fe^2+^ + Zn^2+^)Ratio 1/10High roughnessRatio 2/10Particle morphologyRatio 3/10Dense simple structure[114]GlassZnODCZn^2+^0.0005 mol/dm^3^Dense and very small diameter nanorods(18 nm)0.001 mol/dm^3^Dense and low diameter nanosheets (21 nm)0.005 mol/dm^3^Dense deposits without voids[116]GlassZnODCZn(NO_3_)_2_
30–100 mMWell-defined crystal orientation>100 mMLoss of crystal orientation[117]ITO covered glassZnODCZn(NO_3_)_2_60–90 mMRods with perpendicular orientation and high density>90 mMRods with multidirectional orientation

## 3. Corrosion

Corrosion is a natural phenomenon whereby chemical systems express their tendency towards a state of stable equilibrium. The driving force towards equilibrium is the decrease in free energy, which represents the natural affinity or tendency of the reaction to occur. Any reaction involving a decrease in free energy must not cease, once started, until this parameter has reached a minimum value in the system. This is the case in the corrosion of metallic materials which, with the exception of noble materials, occurs spontaneously and can affect the mechanical properties of the material, compromising the functionality of the metallic device. Zinc electrodeposition coatings have been extensively studied for exhibiting excellent corrosion protection [13,59]. In contact with corrosive environments, zinc oxidises to ZnO, and in the presence of moisture it is converted to hydroxide and, reacting with CO_2_, carbonates [118]. Both compounds are inert, stable and resistant to subsequent exposure to the environment, ensuring a long lifetime for the coating [119]. In addition, when the zinc coating is able to prevent the corrosive environment from reaching the metal surface, we speak of a type of corrosion protection known as a barrier effect [3,120]. This type of protection is characteristic of superhydrophobic coatings [121]. However, it is not always possible to avoid this contact. When the corrosive environment can affect both the coating and the substrate, another type of protection appears, known as zinc cathodic protection [4,120]. In this case, the zinc acts as a sacrificial anode. Sacrificial anodes are highly active metals, with a standard reduction potential that is more electronegative than the material they protect. This potential difference means that the zinc oxidises much faster than the substrate metal [5]. Corrosion will be strongly determined by the texture and microstructure exhibited by the coating surface which, in turn, will depend on the parameters of the electrodeposition process employed [122]. Table 2 shows a summary of the literature on the influence of electrochemical parameters of Zn deposition on the corrosion.

### 3.1. Current Density

Many authors have focused on studying the effect of current density on the corrosion behaviour of Zn electrodeposited coatings. Fazazi et al. [123] investigated the effect of DC density in a range between −16 mA/cm^2^ and −40 mA/cm^2^ on a pure Zn coating. To this purpose, they used the electrochemical impedance spectroscopy technique and the analysis of potentiodynamic polarisation curves to obtain the polarisation resistance (R_p_), the corrosion potential (E_corr_) and the corrosion current density (i_corr_). The Nyquist diagram showed the existence of a semicircle followed by a straight line whose tangent to the real axis was 45°, related to the diffusion of oxygen from the electrolyte into the zinc pores [124,125]. The obtained R_p_ decreased from 36.80 Ω·cm^2^ for −16 mA/cm^2^ to 17.92 Ω·cm^2^ for −32 mA/cm^2^, but it increased again up to 74.26 Ω·cm^2^ for −40 mA/cm^2^. On the other hand, the i_corr_ increased from 89.98 µA/cm^2^ for −16 mA/cm^2^ to 195.59 µA/cm^2^ for −32 mA/cm^2^, with a further decrease to 43.75 µA/cm^2^ for −40 mA/cm^2^. The morphology results showed a compact hexagonal crystal structure typical of Zn deposits at a current density of −16 mA/cm^2^. As the current density increased, between −24 mA/cm^2^ and −32 mA/cm^2^, porous and coarse-grained deposits were obtained. A higher current density of −40 mA/cm^2^ showed a flower-shaped morphology with grain growth in both normal and transverse direction to the surface, generating a transition from two-dimensional to three-dimensional deposit. The authors concluded that the extremely high nucleation rate, induced by the high applied overvoltage and strong hydrogen release, resulted in non-uniform and porous deposits observed when current density was comprised between −24 mA/cm^2^ and −32 mA/cm^2^, causing a higher corrosion rate and then a worsening of the corrosion. However, an intense hydrogen evolution at −40 mA/cm^2^ led to the formation of zinc oxides/hydroxides, with a morphology responsible for the decrease in the corrosion rate and for a better corrosion behaviour [126,127].

The incorporation of other elements to Zn, forming alloy coatings using the electrodeposition technique, has been shown to improve corrosion behaviour [9,82,101,113,128,129,130]. The alloys that have shown the best performance, compared to pure Zn, are those formed with the metals Fe, Co and Ni [131]. Hedge et al. [132] evaluated the corrosion behaviour of electrodeposited ZnFe, ZnNi and ZnNiFe coatings at different current densities on mild steel. Both binary and ternary coatings showed a decrease in i_corr_ with increasing current density. Specifically, for ZnFe and ZnNi, the i_corr_ decreased from 38.8 µA/cm^2^ to 28.7 µA/cm^2^, and from 39.5 µA/cm^2^ to 18.0 µA/cm^2^, respectively, when current density was increased from 10 mA/cm^2^ to 40 mA/cm^2^, but a current density higher than 50 mA/cm^2^ caused a slight increase in i_corr_. A similar behaviour was observed in the ZnNiFe ternary coating, with a decrease in i_corr_ from 8.4 µA/cm^2^ to 2.4 µA/cm^2^ when current density was increased from 10 mA/cm^2^ to 50 mA/cm^2^, and with an increase in the i_corr_ up to 6.8 µA/cm^2^ when current density was 60 mA/cm^2^. Regarding the morphology, the binary coatings showed a dendritic growth, being that the dendrites were smaller for ZnNi, with a Ra roughness value of 35.5 nm and 14.3 nm for ZnFe and ZnNi, respectively. However, the tertiary coating showed a granular morphology with rectangular bars, giving a lower Ra roughness of 6.7 nm. The authors concluded that the smoother granular topography of the ZnNiFe coating allowed for better corrosion behaviour than the binary alloy coatings, possibly due to its significantly higher content of the metals Ni, Fe. Another interesting alloy that has attracted the interest of researchers has been ZnCo. Bhat et al. [133] investigated the corrosion behaviour of the ZnCo coating generated by electrodeposition at different DC densities on low carbon steel. The polarisation curves showed i_corr_ values of 32.32 µA/cm^2^ and 9.021 µA/cm^2^ for an applied current density of 10 mA/cm^2^ and 20 mA/cm^2^, respectively. However, values above 20 mA/cm^2^ resulted in higher i_corr_ up to 28.24 µA/cm^2^ for 50 mA/cm^2^. The coating that exhibited the lowest corrosion rate, 138 µm/year, and thus the best corrosion behaviour, was that obtained with a current density of 20 mA/cm^2^, associated with a lower E_corr_ of −1.081 V and thus a more noble behaviour of the alloy. Interestingly, that alloy showed one of the lowest Co%wt., with a more homogeneous and smoother morphology than the other coverages.

Some authors have also reported that greater grain refinement of coatings implies a better response to corrosion [82,119,130,134,135,136,137,138,139]. It can be expected that a smoothing of the roughness of the coating will avoid the existence of spaces and pores that encourage exposure to the corrosive environment [140,141]. As we have indicated in Section 2.1, the grain size of coatings prepared by PC electrodeposition is smaller than that obtained by DC due to a higher nucleation rate and crystallisation inhibition. Deo et al. [82] confirmed these results for an electrodeposited ZnMn coating on steel, comparing the use of DC and PC. In DC, a current density of 60 mA/cm^2^ resulted in an i_corr_ of 9.39 µA/cm^2^ and a morphology of thin plates with a cauliflower-like appearance. However, the PC current density, irrespective of the operating frequency used, caused an i_corr_ in the range of 3.39–5.83 µA/cm^2^ with a fine-grained, compact and uniform morphology. Kancharla et al. [85] also confirmed these results. They developed an electrodeposited Zn coating on steel by comparing the use of DC and PC. Higher current density values in the range of 10 mA/cm^2^ to 60 mA/cm^2^ generated lower i_corr_ values using DC (from 24.11 µA/cm^2^ to 19.36 µA/cm^2^) than PC (18.95 µA/cm^2^ to 14.93 µA/cm^2^). Moreover, for the highest current density, grain size in line with these results also showed a similar relationship. The maximum current density caused the smallest grain size, with values of 5.73 ± 0.041 µm and 3.89 ± 0.45 µm for DC and PC, respectively, and the lowest corrosion rates, with 0.290 mm/year and 0.223 mm/year for DC and PC, respectively. Other authors wanted to further study the influence of PC density parameters on the corrosion response of a Zn coating. Chandrasekar et al. [6] investigated the effect of PC-density electrodeposition on the corrosion response of a nanocrystalline Zn coating. The electrodeposition was carried out at room temperature and with variations of T_off_ from 6 ms to 60.7 ms, T_on_ from 6 ms to 30 ms and J_p_ from 0.3 mA/cm^2^ to 0.9 mA/cm^2^. It is important to note that when the authors analysed the influence of a specific parameter, they kept the rest of the parameters fixed at values of 60.7 ms, T_off_; 6 ms, T_on_ and 0.5 mA/cm^2^, J_p_. When Toff was increased between values of 18 ms to 30 ms, it generated a flake-like hexagonal crystal morphology. Further increase in T_off_ between 51.5 ms and 60.7 ms caused its higher stacking. According to the authors, this could be due to the formation of the hydroxide anion that blocked the growth of nuclei and enhanced new nucleation. Thus, the higher the T_off_, the thinner the flakes stacked on the substrate and the thinner the thickness of the deposit. This contributed to the improved corrosion performance, as evidenced by the decrease in i_corr_ from 859 A/cm^2^ to 22 A/cm^2^ and the increase in resistance to charge transfer from 7.65 Ω/cm^2^ to 288.78 Ω/cm^2^. When T_on_ was increased between 12 ms to 18 ms, it resulted in a decrease in flake size, but with an increased coating thickness. In this case, there was less nucleation, but more crystal growth. This could contribute to a worse corrosion behaviour of the coating, as evidenced by the increase in i_corr_ from 25 A/cm^2^ to 30 A/cm^2^ with a decrease in resistance to charge transfer from 89.2 Ω/cm^2^ to 70.71 Ω/cm^2^. Finally, the increase in J_p_ from 0.3 mA/cm^2^ to 0.9 mA/cm^2^ caused a decrease in the coating thickness, which is in agreement with the work of Saber et al. [81]. This could be related to an increase in the overpotential, which causes an increase in the free energy to form new nuclei, resulting in a high nucleation rate and a smaller grain size. This resulted in improved corrosion behaviour with a decrease in i_corr_ from 25 A/cm^2^ to 18.5 A/cm^2^. Ultimately, the authors concluded that electrodeposited coatings with a PC density with higher T_off_, J_p_ and lower T_on_ improved corrosion performance.

Another alternative approach to improve the corrosion resistance of pure Zn electrodeposited coatings has been to introduce reinforcing nanoparticles. These are known as composite coatings. The most commonly used nanoparticles have been TiO_2_, Al_2_O_3_, SiO_2_, Fe_2_O_3_, ZrO_2_ and even carbon nanotubes [142,143,144]. The size of the nanostructural materials typically used in electrodeposition is in the range of 1 to 100 nm [145,146,147]. Lofti et al. [14] compared the corrosion behaviour of an electrodeposited ZnNi alloy coating versus an electrodeposited ZnNi/SiO_2_ composite coating. The results indicated that the addition of SiO_2_ nanoparticles led to significant changes in morphology. A significant refinement in grain size and crystal shape took place, which resulted in improved corrosion behaviour. Mehrvarz et al. [148] developed a HA/ZnO composite coating on a NiTi alloy by electrodeposition technique under three different PC densities. Specifically, 3 mA/cm^2^, 6 mA/cm^2^ and 9 mA/cm^2^. Morphological observations by microscopy indicated that a low density of 3 mA/cm^2^ generated a porous coating with non-uniform regions. A higher density of 6 mA/cm^2^ resulted in a compact and uniform surface due to intense nucleation of HA crystals compared to 3 mA/cm^2^. However, the higher density of 9 mA/cm^2^ negatively affected the morphology by causing agglomerations, irregularities and cracks, indicating a lack of adhesion of the coating to the substrate. It is known that an increase in current density in electrodeposition will promote more electrons to leave the anode surface, which will increase the possibility of HA nucleation on the cathode surface. It may also induce an additional overpotential to the electrolyte [149] which would drive the ZnO-like charged nanoparticles to travel faster towards the substrate, increasing the code position of HA crystals and ZnO nanoparticles on the surface. However, this does not happen. According to the literature, the isoelectric point of Zn is in the range of 9 to 10. When ZnO nanoparticles are immersed in an aqueous environment, they form a hydroxide layer on their surface that can acquire a positive charge if the pH of the solution is below the isoelectric point and vice versa [150]. However, a high current density can lead to an increase in pH in the vicinity of the cathode surface due to excessive hydrolysis of water [151]. Perhaps this weakened the surface charge of the nanoparticles, which near the isoelectric point can render them neutral [152]. Potentiodynamic polarisation curves showed that coatings carried out at current densities of 3 mA/cm^2^ and 9 mA/cm^2^ showed high i_corr_ values of 17.9 nA/cm^2^ and 32.9 nA/cm^2^ and low polarisation resistances of 1.29 MΩ and 0.69 MΩ, respectively. However, a density of 6 mA/cm^2^ exhibited the lowest i_corr_, with a value of 5.7 nA/cm^2^ and the highest polarisation resistance of 4.29 MΩ. This corrosion behaviour agrees with the morphological results described above. The authors concluded that the electrodeposited surface with a density of 6 mA/cm^2^ was the densest and most compact with HA crystals and exhibited the best corrosion behaviour. Sajjadnejad et al. [153] developed Zn/TiO_2_ composite coatings at a nanoparticle concentration of 5 g/L by electrodeposition technique with different DC densities. Increasing the density from 0.08 A/cm^2^ to 0.12 A/cm^2^ resulted in the appearance of a very fine microstructure coating with a very uniform TiO_2_ nanoparticle morphology throughout the coating. The incorporation of nanoparticles into the coating, in addition to providing new nucleation sites, was also able to disrupt the crystalline growth of the deposit [154]. In general, the Zn/TiO_2_ coating usually has hexagonal grains oriented perpendicular to the surface, resulting in a surface with numerous defects. However, under an overpotential on the cathode surface, the Zn deposit also grows in the direction normal to the surface [155], generating more compact surfaces. This behaviour is evident in the response of the coating to corrosion. The lowest i_corr_ and corrosion rate values of 2.70 µA/cm^2^ and 0.031 mm/year were obtained for the higher density of 0.12 A/cm^2^. In contrast, the highest i_corr_ and corrosion rate values of 265 µA/cm^2^ and 3.084 mm/year were obtained for the lower density of 0.08 A/cm^2^. These results are in agreement with those obtained by Cabral-Miramontes et al. [72] in the development of a Zn/TiO_2_ composite coating at DC densities of 50 mA/cm^2^ and 100 mA/cm^2^. Higher current density generated a lower i_corr_ of 1.4 µA/cm^2^ compared to 9.0 µA/cm^2^ at a lower current density. These results can be explained with those obtained in their morphology. Higher current density promoted grain size refinement and higher compactness of the composite coating. Al-Dhire et al. [135] also investigated the effect of DC density on the deposition of Zn/SiC composite coating on steel. The electrodeposition was carried out in a ZnSO_4_ bath with a concentration of 20 g/L SiC. The coating obtained at a low density of 20 mA/cm^2^ showed a large hexagonal grain morphology perpendicular to the substrate, with few SiC particles detected in the Zn matrix. However, as the current density increased from 20 mA/cm^2^ to 40 mA/cm^2^, so did the SiC particle embedding, resulting in a mixed structure of large and small grains. A similar observation was described by Sajjadnejad et al. [153]. Regarding the corrosion behaviour of the coating, a low value of 20 mA/cm^2^ generated a high value of i_corr_ and corrosion rates of 0.264 A/cm^2^ and 0.39 mm/year, respectively. This result is in accordance with the exhibited microstructure of a hexagonal structure perpendicular to the substrate with hollow spaces. A higher value between 20 mA/cm^2^ to 40 mA/cm^2^ caused a decrease in i_corr_ and corrosion rates of 0.209 A/cm^2^ and 0.3 mm/year, respectively. This is typical of the morphology exhibited with smaller grains due to the higher embedding of SiC particles. This improved corrosion behaviour is attributed to the formation of corrosion microcells formed by the SiC particles in the Zn layer. In these cells, the SiC particles act as cathodes, while the Zn matrix acts as an anode. The potential of SiC is more positive than that of Zn, which could facilitate anodic polarisation and result in the inhibition of localised corrosion [156]. Finally, a higher current density of 50 mA/cm^2^ caused a slight increase in the i_corr_ and corrosion rates of 0.2097 A/cm^2^ and 3.1 mm/year, respectively. This could be due to the lower concentration of SiC nanoparticles in the Zn coating at this current density. This effect was also observed and described by Mehrvarz et al. [148]. Other authors have also studied the combined action of various nanoparticles in the electrodeposited Zn matrix. This is the case of Daniyan et al. [157], who investigated the effect of DC current density on the microstructure and mechanical properties of Zn/TiO_2_ and Zn/TiO_2_-WO_3_ coatings on carbon steel. The electrolytic bath used contained chlorides and a concentration of 20 g/L TiO_2_ and 15 g/L WO_3_ at 400 rpm agitation. When the electrodeposition took place at a current density higher than 830 A/cm^2^, the composite coatings showed higher thicknesses compared to a lower current density of 560 A/cm^2^. The combined incorporation of WO_3_ and TiO_2_ generated a refined morphology with finer nodular particles compared to the coating without WO_3_ nanoparticles. These nodular structures were more pronounced at higher current densities. This led to a decrease in the i_corr_ and corrosion rate. The authors concluded that this combined coating of TiO_2_-WO_3_ nanoparticles on the electrodeposited Zn matrix exhibited excellent corrosion properties. In conclusion, the presence of nanoparticles improves the corrosion resistance of the metal matrix in four ways. First, the presence of the nanoparticles leads to the formation of microcells in the zinc matrix. In these microcells, the zinc acts as an anode and the TiO_2_ nanoparticles act as a cathode, which facilitates anodic polarisation, and consequently only homogeneous corrosion occurs. Secondly, the evenly distributed nanoparticles act as physical barriers to the initiation and growth of corrosion defects. Third, the nanoparticles are reported to fill the voids and defects between the grains, resulting in a smoother surface and improved corrosion resistance. And fourth, the incorporation of nanoparticles reduces the metal area available for corrosion [156,158].

### 3.2. Temperature

Temperature plays an important role in controlling the corrosion behaviour of the coating [159,160]. The increase in temperature will influence the deposition rate and grain size of the Zn crystal due to changes in nucleation density [13].

Jiang et al. [161] investigated the influence of different temperatures on the corrosion behaviour of a Zn coating on a Cu substrate. They employed the DC electrodeposition technique with a current density of 5 mA/cm^2^ and a temperature in the range between 50 °C and 70 °C. A low electrodeposition temperature of 50 °C generated a smooth morphology of fine grain flakes, but with small pores between the particles. It is known that the presence of pores facilitates the contact between the electrolyte and the Zn coating, leading to an increase in the i_corr_ [162]. For this reason, the coating obtained at low temperature exhibited the worst corrosion behaviour with an icorr value of 37.69 µA/cm^2^. Increasing the temperature to 60 °C showed improved coating compactness and pore elimination. This resulted in a decrease in the i_corr_ value to 5.18 µA/cm^2^, exhibiting the best corrosion behaviour of the coating. A further increase at 70 °C contributed to improving the reaction rate and promoted the growth of the nuclei, leading to the formation of even larger and rougher grain flakes. This resulted in an increase in the i_corr_ to 13.69 µA/cm^2^ and a worsening of the corrosion behaviour. Similar behaviour was reported by Chen et al. [163] who investigated the influence of different temperatures on the corrosion behaviour of a Zn coating on mild steel. For this purpose, they used a DC density of 4 mA/cm^2^ and temperatures ranging from 28 °C to 80 °C. At low temperatures between 28 °C and 40 °C, the morphology of the coating was flat. They observed that the grain density and compactness increased with temperature. This resulted in a decrease in i_corr_ from 42.9 µA/cm^2^ to 17.4 µA/cm^2^ and an increase in R_p_ from 253 Ω·cm^2^ to 621.90 Ω·cm^2^. This led to a better response to corrosion. Higher temperatures between 60 °C and 80 °C generated a staggered structure. This is because at low temperature the Zn flake-like grains tend to grow in a direction parallel to the substrate. Hence, they show a flat morphology. On the other hand, a higher temperature caused the Zn flakes to grow in a direction with a higher angle relative to the substrate [164]. At 60 °C the coating showed the best compactness and lowest roughness. This phenomenon is associated with an increase in deposit efficiency and a drop in energy consumption. This suggested a growth of nucleation and an inhibition of the crystallisation process of the Zn grains. This resulted in a decrease in the i_corr_ to 10.9 µA/cm^2^ and an increase in the R_p_ of 1053.11 Ω·cm^2^. This temperature value was proposed by the authors as the optimum in corrosion behaviour and coincides with that proposed in the work of Jiang et al. [161]. An increase in the temperature to 80 °C was so high that it led to the decomposition of the electrolyte. Bubbles started to appear during the electrodeposition process and resulted in a non-compact coating. This led to a worsening of the corrosion with an i_corr_ of 41.5 µA/cm^2^ and an R_p_ of 376 Ω·cm^2^. Moreover, Saidi et al. [95] described this behaviour in the development of a ZnO coating on Ti6Al4V at different temperatures. At a low temperature of 40 °C, the coating showed a spherical-like morphology with very low uniformity. An increase in temperature to 55 °C caused the spheres to melt into a worm-like structure. A temperature higher than 70 °C also caused a porous and irregular flower-shaped morphology with increased roughness. The corrosion behaviour was shown to be detrimental when the electrodeposited coating was fabricated at a temperature of 70 °C. A similar result at low temperatures was reported by Zhang et al. [88]. The authors evaluated the effect of temperature in the range of 30 °C and 45 °C on the corrosion behaviour of a Zn deposit on pure Al foil. An increase in the electrodeposition temperature resulted in a progressive decrease in the Zn grain size, leading to a more compact surface. This resulted in a decrease in the i_corr_ and an improvement of the corrosion behaviour. In short, there seems to be a consensus in the literature that electrodeposition temperatures close to 60 °C improve the corrosion performance of the Zn coating. However, temperatures farther away from this temperature, both above and below it, lead to coatings with worse corrosion performance.

### 3.3. Additives

The use of additives in the electrolytic bath during the electrodeposition process will influence the corrosion behaviour of the deposit. This is because the additives inhibit grain growth and condition its directionality. This eliminates possible over stresses. Moreover, they prevent the formation of hydrogen bubbles that can end up diffusing inside the substrate and generating possible internal stresses [165]. Ultimately, they encourage the formation of more compact, homogeneous and brighter deposits.

Trejo et al. [111] investigated the influence of different polyethoxylated additives on the anticorrosive properties of a Zn coating from an acid chloride electrolyte. The additives used were ethylene glycol polymers of different molecular masses: 400 g/mol, 8000 g/mol and 20,000 g/mol. The electrodeposition technique they used was with a DC density. In the absence of additives, the morphology of the coating was in the form of hexagonal plates, typical of pure Zn deposits with large grain sizes [166,167]. This caused an instantaneous nucleation mechanism. This morphology encouraged the highest i_corr_ of all the surfaces studied, with a value of 2.39 µA/cm^2^ and the worst corrosion rate of 0.036 mm/year. In the presence of additives, the deposits presented a morphology of scales grouped in clusters constituting a nodular structure with a smaller grain size, specifically of 14 µm^2^ and 22 µm^2^ for additive molecular masses of 400 g/mol and 8000 g/mol, respectively. This originated a better behaviour against corrosion, with i_corr_ of 1.88 µA/cm^2^ and 0.28 µA/cm^2^ and corrosion rates of 0.028 mm/year and 0.004 mm/year. However, the inclusion of the additive with the highest molecular mass (20,000 g/mol) caused an increase in grain size. This morphology impaired the corrosion behaviour of the coating, with an i_corr_ of 1.07 µA/cm^2^ and a corrosion rate of 0.016 mm/year. This is possibly due to a blocking effect caused by the additive, referring to a reduction in both the number of active sites and the nucleation rate. This effect was also reported by Michailova et al. [168,169] in the electrodeposition of copper in the presence of a polyethoxylated organic additive. Similar work was carried out by Zaabar et al. [170]. They fabricated a Zn coating on carbon steel from an acid sulphate electrolyte containing nettle extract as an additive. The concentration of additive used was 1.5 g/L and the electrostatic potential used was −1700 mV in DC mode at room temperature. Without additive, the morphology of the electrodeposited Zn coating showed large aggregates of closely packed hexagonal Zn crystals, with a preferential growth orientation perpendicular to the substrate and with a relatively high roughness. This morphology generated a bad behaviour against corrosion. The inclusion of nettle extract in the electrolyte caused a modification in the morphology of the Zn crystal. The Zn grains were transformed into fine crystals of submicronic, almost nanometre sizes, with a growth orientation parallel to the substrate. This caused the coating to be very uniform and compact, generating a coating with a high resistance to corrosion. Khorsand et al. [8] also investigated the influence of different additives on the anticorrosion behaviour of a Zn coating on steel. They used 10 ppm of Sn^2+^ or 0.5 M oxalic acid as additives. The DC density used was 100 mA/cm^2^ and 400 mA/cm^2^ for Sn^2+^ and oxalic acid, respectively. In the absence of additives, the Zn coating showed a typical hexagonal platelet morphology with a multidirectional orientation. This denoted a greater number of imperfections. This coating showed the worst behaviour against corrosion. The addition of Sn cations caused growth with preferential direction of Zn crystals, consisting of numerous separately stacked hexagonal platelets parallel to the substrate. On the contrary, the addition of oxalate anions modified the morphology to a granular nature, free of imperfections, compact and with less roughness. This caused the best behaviour against corrosion. These results are in agreement with other authors, who reported that the adsorption of additives on the cathode surface retarded grain growth but increased the nucleation rate, giving rise to finer and more compact grain deposits [8,171,172,173]. These compact and therefore less rough coatings had a lower corrosion rate as a consequence of the exposure of a smaller surface area to the corrosive medium. This effect was also observed with other additives such as gelatine, polyethylene glycol, saccharin, tetrabutylammonium chloride and sodium lauryl sulphate [174,175,176].

The use of additives in combination with PC electrodeposition has been shown to achieve a much more effective grain refinement in the nanometric order. Nanocrystalline materials produce a high volume fraction at grain boundaries that can produce better corrosion performance compared to coarse-grained materials [124,177]. This is because the metal atoms found at the grain boundaries have a higher activity and are prone to corrosion. The grain boundary volume fraction increases with the nanocrystallization of the electrodeposited surface, thus the number of surface activity sites increases, causing the nanocrystalline Zn coating surface to quickly form a protective film of corrosion products compared to a thick crystalline Zn coating [178]. In this sense, Chandrasekar et al. [6] compared the response to the corrosion of a monocrystalline Zn coating obtained by PC density, with and without additive. The primary additive used was polyvinyl alcohol (PVA) and the secondary was piperonal. The fixed parameters of the electrodeposition were 60.7 ms, T_off_; 6 ms, T_on_ and 0.5 mA/cm^2^, J_p_. When electrodeposition was carried out without additives the dimension of the flake crystals was approximately 1.45 µm with an RMS roughness of 530 nm. This generated an i_corr_ of ~18.5 A/cm^2^. The inclusion of the PVA primary additive at a concentration of 0.9 g/L modified the morphology of the deposit, with more refined grains of 50 nm and a smoother RMS roughness of 350 nm. The inclusion of the secondary additive of 0.9 g/L of PVA + 0.4 g/L of piperonal caused an even greater refinement of the grain, with a size of 33 nm and RMS roughness of 200 nm. The authors concluded that the addition of additives in density of PC caused a spectacular grain refinement that resulted in a better behaviour against corrosion, with an i_corr_ value of 16 A/cm^2^, much lower than the behaviour without additives. The compact, adherent and homogeneous structure was the plausible explanation to justify the low value of the i_corr_. Higher concentrations of additives caused irregular, porous topographies and poorer behaviour against corrosion. Moreover, Li et al. [178] developed a nanocrystalline Zn coating by the combined use of double-pulsed PC density and polyacrylamide additive in an acid sulphate electrolyte. The additive concentrations used varied from 0.5 g/L to 2 g/L. Electrodeposition was performed by varying the forward peak current density, J_pd_, from 0.1 mA/cm^2^ to 0.4 mA/cm^2^ and the reverse peak current density, J_pi_, from 0.02 mA/cm^2^ to 0.05 mA/cm^2^. Throughout the study, the T_on_ and T_off_ values were kept constant at 0.2 ms and 0.8 ms, respectively. Without the use of additive, the coating showed a typical Zn grain morphology, shaped like a hexagonal plate aligned with a preferred orientation parallel to the substrate. Increasing J_pd_ from 0.1 mA/cm^2^ to 0.7 mA/cm^2^ caused an increase in grain size. This result indicated that the unique modification of the electrodeposition parameters would not allow to obtain Zn nanocrystals. The smallest grain size was obtained for a J_pd_ of 0.1 mA/cm^2^. For this reason, this value was kept fixed to study the influence of the different concentrations of additives. The corrosion behaviour of the coating without additive was not good. Morphologically, it appeared that the coating had melted. The addition of the additive in a concentration of 1 g/L caused a smaller grain size with a relatively softer and more uniform Zn crystal. As the concentration continued to increase the coatings thickened and the additive precipitated. The influence of different J_pd_ in the range of 0.1 mA/cm^2^ to 0.4 mA/cm^2^ was examined, keeping the additive concentration fixed at 1 g/L. The increase in J_pd_ between 0.1 mA/cm^2^ to 0.3 mA/cm^2^ caused a decrease in grain size. This can be explained by Sherik’s theory [179]. Increased PC density can cause an overpotential to appear, which increases the free energy to form new nuclei and results in a higher nucleation rate and smaller grain size. The smallest grain size was obtained for a J_pd_ of 0.3 mA/cm^2^ and an additive concentration of 1 g/L with a value of 42 nm. However, for J_pd_ of 0.4 mA/cm^2^ it caused the grain size to become relatively coarse. The influence of J_pi_ from 0.02 mA/cm^2^ to 0.05 mA/cm^2^ was also studied, keeping the concentration of the additive constant. However, no clear effect was detected. This coating morphology obtained by electrodeposition with additive and J_pd_ of 0.3 mA/cm^2^, resulted in excellent corrosion resistance.

### 3.4. Electroactive Ions

The concentration of electroactive ions contained in the electrolyte during the Zn electrodeposition process is a parameter that will condition its behaviour against corrosion. We have already described that changes in the concentration would cause morphological modifications related to the size, density and orientation of the Zn grains; in short, changes that will modify the compactness, uniformity and thickness of the coating that directly affect the corrosive capabilities of the coating. Abedini et al. [180] studied the influence of different concentrations of Mn^2+^ ions (between 1.14 µmol/L and 2.28 µmol/L) in alkaline solution on the anticorrosive behaviour of a ZnNiMn alloy coating. The increase in the concentration of Mn^2+^ significantly affected the morphology, chemical composition and anticorrosive capacity of the coating. A low concentration of 1.14 µmol/L of Mn^2+^ ions generated the appearance of U-shaped structures, surrounded by a large number of small globular nodules, with non-uniform distribution and with microcracks, both along the entire surface and in the nodules. At this concentration, approximately a 6%wt. of Mn and 13%wt. of Ni was detected. However, increasing the concentration of Mn^2+^ ions to 2.28 µmol/L caused a smoother morphology without U-shaped structures and with a significant decrease in globular nodules. In this case, approximately an 8.5%wt. of Mn and 9.5%wt. of Ni were detected. It is known that the addition of Ni to the Zn coating changes the E_corr_ to more noble potentials, while adding Mn to the ZnNi coating changes the E_corr_ to more negative potentials, due to its less noble nature. Based on this, as the authors detected an increase in E_corr_ from −848 mV to −1358 mV, and they were able to directly associate it with a higher concentration of Mn in the coating. The increase in the concentration of Mn^2+^ ions in the electrolyte caused a better behaviour against corrosion. There was a decrease in i_corr_ from 457 µA/cm^2^ to 30.65 µA/cm^2^ and an increase in Rp from 138 Ω·cm^2^ to 1065 Ω·cm^2^. The authors attributed this to the formation of a blemish-free surface, a smaller grain size and the generated corrosion product of MnO_2_, which achieved excellent protective properties in the coating.

Fayomi et al. [181] investigated the influence of different concentrations of ZnO nanoparticles on the corrosion response of a Zn/ZnO composite coating on carbon steel. The concentrations of ZnO nanoparticles used were 20 g/L and 40 g/L. The electrodeposition parameters of 0.5 A/cm^2^ (DC) and 40 °C were also used. A low concentration of 20 g/L generated a morphology of hexagonal crystals with a large grain size. In this case, a high value of i_corr_ and a low R_p_ were recorded, with a value of 2310 mA/cm^2^ and 80,705 Ω, respectively. A high concentration of 40 g/L showed a coating with good adherence, fine grain size and uniform distribution of nanoparticles in the Zn matrix. This originated a low value i_corr_ of 1.166 mA/cm^2^ and a high R_p_ value of 291.38 Ω. It is known that the incorporation of nanoparticles in a metallic matrix promotes the increase in the number of nucleation sites and prevents crystal growth, resulting in smaller grain sizes [182]. A higher concentration could favour this trend. In short, higher concentrations of ZnO nanoparticles generated better behaviour against corrosion. However, this behaviour is not extensible for all nanoparticles embedded in the Zn matrix. Malatji et al. [183] studied the corrosion behaviour of Zn/Cr_2_O_3_ and Zn/SiO_2_ composite coatings on mild steel at different concentrations of nanoparticles. The concentrations used were 10 g/L and 20 g/L for Cr_2_O_3_, and 5 g/L and 10 g/L for SiO_2_. The electrochemical parameters used were 1.5 A (DC) and 25 °C. The unencrusted materials showed a typical morphology of the pure Zn electrodeposited coating with a flake-like crystal structure. Unlike this, however, the composite coatings exhibited a more compact microstructure and smaller grain size. The compact coatings obtained with Cr_2_O_3_ nanoparticles showed a better microstructure compared to those obtained with SiO_2_ nanoparticles. This was justified by the authors due to the finer nature and tendency towards agglomeration that the SiO_2_ nanoparticles showed. Furthermore, SiO_2_ nanoparticles are hydrophilic and their codeposition on the cathode surface is difficult [184]. Some authors attribute the insufficient codeposition of these particles to the generation of a poor coating quality [185]. On the other hand, the Cr_2_O_3_ particles could be more easily dispersed in the electrolyte, facilitating their incorporation into the Zn matrix. The results showed an incorporation in the Zn matrix of 3.8%wt. for Cr_2_O_3_ and 0.85%wt. for SiO_2_. The increase in the concentration of SiO_2_ nanoparticles in the electrolyte encouraged the agglomeration of the nanoparticles. This caused the non-existence of significant changes in the value of E_corr_ and, therefore, no improvement against corrosion. On the other hand, the increase in the Cr_2_O_3_ concentration yielded positive results against corrosion. The E_corr_ went from a value of −1.1972 to −1.1075 V. Hamid et al. [186] obtained similar results with Cr_2_O_3_ nanoparticles. They reported that the inclusion of these nanoparticles in the Zn matrix also acted as a corrosion inhibitor. A similar result was reported by Blejan et al. [185]. They investigated the anticorrosion properties of a ZnNi/Al_2_O_3_ composite coating at different concentrations of Al_2_O_3_ nanoparticles on carbon steel. The concentrations tested were between 5 g/L and 15 g/L. The electrodeposition parameters were 20 mA/cm^2^ (DC) and 23 °C. The increase in the concentration of nanoparticles modified the deposit towards a finer grain because the nanoparticles intervened in the nucleation growth process. This caused a worsening of the corrosion. Specifically, an increase in i_corr_ and a decrease in Rp were obtained, with values of 1.23 µA/cm^2^ to 2.57 µA/cm^2^ and 4024.9 Ω·cm^2^ to 1190 Ω·cm^2^, respectively. The authors stated that they did not know the exact causes of this change in behaviour. However, a similar result was described in the works of Kondo et al. [187] and Malatji et al. [183] for SiO_2_ nanoparticles. These authors stated that not only the SiO_2_ nanoparticles would suffer agglomerations in the electrolyte that could impair their correct dispersion in the Zn matrix, resulting in morphological imperfections. Al_2_O_3_ nanoparticles also suffered from it. Similarly, He et al. [113] detected a worse behaviour against corrosion with the increase in the concentration of Zn^2+^ ions in the electrolytic bath for the electrodeposition of a ZnFe alloy. In this case, the authors associated it with the %wt. of the alloying elements. A higher concentration of Zn^2+^ ions in the electrolyte caused a higher %wt. of Zn and lower %wt. of Fe in the electrodeposited coating. As Fe is a metal of greater nobility than Zn, its smaller presence gave it a worse resistance against corrosion.
materials-16-05985-t002_Table 2Table 2Summary of the literature on the influence of electrochemical parameters on corrosion.Ref.SurfaceCoatingCurrentTypeCurrent Density (mA/cm^2^)Corrosion[123]Steel Pure ZnDC−16 i_corr_ 89.98 µA/cm^2^R_p_ 36.80 Ω·cm^2^−32 i_corr_ 195.59 µA/cm^2^R_p_ 17.92 Ω·cm^2^−40 i_corr_ 43.75 µA/cm^2^R_p_ 74.26 Ω·cm^2^[132]Mild steelZnFeDC10–60i_corr_ decreases with increasing current density but increases slightly when values equal to or greater than 50 mA/cm^2^ are reachedZnNiZnNiFe[133]Carbon steelZnCoDC10 i_corr_ 32.32 µA/cm^2^20icorr 9.021 µA/cm^2^50i_corr_ 28.24 µA/cm^2^[82]SteelZnMnDC60 i_corr_ 9.39 µA/cm^2^PC(T_on_ 12–10.6 ms and T_off_ 28–2.6 ms)i_corr_ 3.39–5.83 µA/cm^2^[85]SteelPure Zn DC10–60i_corr_ 24.11–19.36 µA/cm^2^PC(T_on_ 60 ms and T_off_ 240 ms)i_corr_ 18.95–14.93 µA/cm^2^[6]Mild steelPure ZnPC(T_on_ 6 ms and T_off_ 60.7 ms)0.5The increase in J_p_ from 0.3 to 0.9 mA/cm^2^ causes a decrease in i_corr_ from 25 from 18.5 A/cm^2^The increase in Ton from 12 to 18 ms causes an increase in i_corr_ from 25 from 30 A/cm^2^The increase in T_off_ from 18 to 60.7 ms causes a decrease in i_corr_ from 859 to 22 A/cm^2^[148]NiTi HA/ZnO DC3 i_corr_ 17.9 nA/cm^2^Rp 1.29 MΩ6 i_corr_ 5.7 nA/cm^2^Rp 4.29 MΩ9 i_corr_ 32.9 nA/cm^2^Rp 0.69 MΩ[153]SteelZn/TiO_2_
DC80i_corr_ 265 µA/cm^2^120i_corr_ 2.70 µA/cm^2^[72]1018 carbon steelZn/TiO_2_DC50i_corr_ 9.0 µA/cm^2^100i_corr_ 1.4 µA/cm^2^[135]SteelZn/SiCDC20i_corr_ 0.264 A/cm^2^40i_corr_ 0.209 A/cm^2^50i_corr_ 0.2097 A/cm^2^[157]Carbon steelZn/TiO_2_-WO_3_DC830 × 10^3^The incorporation of TiO_2_-WO_3_ leads to a decrease in i_corr_560 × 10^3^**Ref.****Surface****Coating****Current Type****Temperature****(°C)****Corrosion**[161]CopperPure ZnDC50i_corr_ 37.69 µA/cm^2^60i_corr_ 5.18 µA/cm^2^70i_corr_ 13.69 µA/cm^2^[163]Mild steelPure ZnDC28i_corr_ 42.9 µA/cm^2^R_p_ 253 Ω·cm^2^40i_corr_ 17.4 µA/cm^2^R_p_ 621.90 Ω·cm^2^60i_corr_ 10.9 µA/cm^2^R_p_ 1053.11 Ω·cm^2^80i_corr_ 41.5 µA/cm^2^R_p_ 376 Ω·cm^2^[88]AluminumPure ZnDC30The increase in temperature causes a decrease in i_corr_45**Ref.****Surface****Coating****Current****Type****Additives****Corrosion**[111]Glassy carbonPure ZnDCEthylene glycol400 g/moli_corr_ 1.88 µA/cm^2^800 g/moli_corr_ 0.28 µA/cm^2^20,000 g/moli_corr_ 1.07 µA/cm^2^[6] Mild steelPure ZnPC(J_p_ 0.5 mA/cm^2^, T_on_ 6 ms and T_off_ 60.7 ms)PVA (0.9 g/L)+ Piperonal (0.4 g/L)Without additives causes an i_corr_ 18.5 A/cm^2^. The adition of both additives causes an i_corr_ 16 A/cm^2^[178]CooperPure ZnPC(T_on_ 0.2 ms and T_off_ 0.8 ms)Polyacrylamide (1 g/L)The corrosion behaviour without additive was not good. With additive and J_pd_ of 0.3 mA/cm^2^ excellent corrosion resistance. No clear effect was detected with J_pi_**Ref.****Surface****Coating****Current****Type****Electroactive Ions****Corrosion**[180]Glassy carbonZnNiMnDCMn^2+^1.14 µmol/Li_corr_ 457 µA/cm^2^R_p_ 138 Ω·cm^2^2.28 µmol/Li_corr_ 30.65 µA/cm^2^R_p_ 1065 Ω·cm^2^[181]Carbon steelZn/ZnODCZn nanoparticles20 g/Li_corr_ 2310 mA/cm^2^R_p_ 80.705 Ω40 g/Li_corr_ 1.166 mA/cm^2^R_p_ 291.38 Ω[183]Mild steelZn/Cr_2_O_3_DCCr_2_O_3_10 g/LE_corr_ − 1.1972 V20 g/LE_corr_ − 1.1075 VZn/SiO_2_SiO_2_5 g/LHigher concentration does not cause improvement in corrosion10 g/L[185]Carbon SteelZnNi/Al_2_O_3_DCAl_2_O_3_ nanoparticles5 g/Li_corr_ 1.23 µA/cm^2^R_p_ 4024.9 Ω· cm^2^15 g/Li_corr_ 2.57 µA/cm^2^R_p_ 1190 Ω· cm^2^

## 4. Antimicrobial Behaviour of Zn Coatings

Zinc is a promising ally for the protection of biomaterials against the bacterial invasion of their surface [188,189,190,191]. It is known that an excess of metal ions can be toxic to bacterial cells [46]. Hence, there is some threshold of concentration of Zn^2+^ ions necessary to hamper bacterial metabolic functions [41,46]. Reddy et al. [38] reported that *Escherichia coli* (*E. coli*) took advantage of low Zn^2+^ concentrations as nutrients, favouring their growth. However, when higher ion concentrations were in media, growth was inhibited. Inside bacteria, Zn^2+^ ions interact with nucleic acids and inactivate respiratory system enzymes [38,192], causing cell death. This excess of Zn^2+^ into the cell comes from the deregulation of zinc homeostasis, which permeates the bacterial membrane, allowing an extra entry of ions [193]. Consequently, the concentration of Zn^2+^ in media is relevant for its antibacterial behaviour. For a material, Pasquet et al. [41] determined that the release mechanism of Zn^2+^ in the medium is affected by two main groups of parameters: (a) the physicochemical properties such as morphology, size, concentration and porosity, and (b) the chemistry of the dissolution media, UV illumination, exposure time and the presence of other elements. However, the influence of these parameters is not entirely clear [194]. Table 3 shows a summary of the literature on the influence of electrochemical parameters of Zn deposition on antibacterial behaviour.

In the search for protective coatings against bacterial colonization, pure Zn coatings manufactured by electrodeposition have generated interesting results. Dawari et al. [195] fabricated electrodeposited Zn coatings on flat and smooth plates and on micropatterned plates of tungsten–cobalt. The electrodeposition technique was carried out with the DC technique and the strain selected for the microbial study was *Staphylococcus aureus* (*S. aureus*). However, the plates were previously worn before using. For the test, bacteria were placed in contact with both coatings during an incubation time of 24 h. By direct contact, they were transferred to an agar plate to be incubated and later the colonies were counted. The authors observed that micropatterned surfaces completely inhibited the growth of *S. aureus*, but they did not observe the inhibition on the flat and smooth surfaces. The authors indicated that the previous wear destroyed practically the entire Zn coating of the flat surface. Nevertheless, on the micropatterned surface, Zn deposits on only the upper part of the pillars was removed, keeping the coating between the micropillars protected, with a sufficient concentration of Zn to cause bacterial death. Kultamaa et al. [196] also developed a pure Zn coating electrodeposited on a 316 L stainless steel surface where holes were previously produced, and they evaluated its antimicrobial activity against *S. aureus*. The morphology of the electrocoating was of flake-shaped crystals. A polishing treatment allowed the coating to be kept only inside the pores but removed from the rest of the substrate. Antimicrobial studies confirmed that the coating within the pores was sufficient to completely inhibit the growth of *S. aureus*. According to the authors, it was possible to achieve effective antibacterial activity using significantly smaller amounts than those needed for a complete surface coating.

Zn alloys embedded with electrodeposited particles have also been studied for action against microorganisms. Momeni et al. [191] electrodeposited a ZnNi alloy composite coating embedded with TiO_2_ nanoparticles. The antibacterial activity was evaluated by the agar diffusion method against *S. aureus* (Gram-positive) and *E. coli* (Gram-negative). The ZnNi coating without TiO_2_ embedment showed an inhibition region of 18.5 mm and 25 mm for *E. coli* and *S. aureus*, respectively. However, the embedding of TiO_2_ nanoparticles in the ZnNi matrix at a concentration of 3 g/L improved the antimicrobial behaviour up to an inhibition zone of 23 mm and 28 mm for *E. coli* and *S. aureus*, respectively. Higher activity against *S. aureus* than *E. coli* was attributed to the structural and compositional differences of the cell membranes of both strains. Although they have similar internal structures, Gram-positive bacteria have a relatively thick, covalently bound peptidoglycan outer plasma membrane containing teichoic and lipoteichoic acids, while Gram-negative bacteria have a thin layer of peptidoglycan and an outer membrane containing lipopolysaccharides, phospholipids and proteins [197,198], and are even sensitive to Zn^2+^ ions [199]. In addition, as the surfaces of most microorganisms are negatively charged, and Zn coatings will transfer a positive charge in the form of Zn^2+^, an electrical attraction between the Zn particles and the bacteria appears. In this situation, as with other antimicrobials [200], the permeability of the bacterial wall increases and the uncontrolled mass transport of these cations may cause destabilization and obstruction of mass transport, ultimately triggering bacterial death [201]. Gopi et al. [130] electrodeposited an HA coating with Mg and Zn embedded on Ti6Al4V alloy, by the PC method, at two conditions (condition 1: J_p_ = 1.0 mA/cm^2^, T_on_ = 1 s and T_off_ = 4 s, and condition 2: J_p_ = 1.0 mA/cm^2^, T_on_ = 4 s and Toff = 1 s)., The antimicrobial activity was evaluated for the strains *S. aureus* and *E. coli* by the agar diffusion method. Condition 1 exhibited a morphology of spherical and compact particles, with a relatively high content of Zn and Mg, leading to a good behaviour against corrosion and intense antimicrobial activity against both strains. However, condition 2 caused a morphology with flakes with a multitude of aggregates and a very low content of Zn and Mg, resulting in worse corrosion performance and antimicrobial activity than condition 1. Nevertheless, both conditions exhibited better corrosion performance and antimicrobial activity than the uncoated Ti6Al4V surface. Furthermore, *E. coli* was shown to be less susceptible to the coating than *S. aureus*, as in the Momeni et al. research [191]. Zhai et al. [202] also evaluated the antimicrobial activity of a ZnNi alloy coating and a ZnNi/chitosan composite coating on carbon steel. Both coatings were electrodeposited according to the DC method. Their antibacterial activity against *E. coli* was tested by direct contact of surfaces with a suspension of 10^6^ CFU/mL of *E. coli* for 24 h. After that time, authors found that the adhered bacteria on the ZnNi coating was 2.92% of surface, but 0,18% on the ZnNi/chitosan composite coating.

Rationale of the antibacterial action of ZnO electrodeposited coatings has also been reported in the literature. It has been shown to have various mechanisms of action against microorganisms, including the release of Zn^2+^ in the physiological environment [203,204,205], the physical alterations produced in the plasmatic membrane and the generation of reactive oxygen species (ROS) [204]. Its effectiveness will depend on the morphology of the electrodeposited coating, the grain size [204,206,207] as well as the binding of ZnO particles to cells by electrostatic interactions [208,209]. Zhai et al. [190] fabricated an electrodeposited coating of ZnO on the surface of pure Zn. Their goal was to achieve a marine antifouling material with antibacterial activity against *E. coli*. The electrodeposition was carried out under a DC procedure and at different concentrations of the capsaicin additive (0.2 g/L and 0.6 g/L). The coating without additive showed a hexagonal crystal structure which is typical for electrodeposited Zn, a low capacity against corrosion and a coverage of *E. coli* of 1.5%, but bacterial viability was intact. However, the addition of 0.4 g/L of capsaicin in the electrolyte generated a morphology of ZnO nanopillars, an improved corrosion behaviour and a reduction in *E. coli* coverage of 99.9% and an excellent antibacterial behaviour of 99.96%. Shyu et al. [210] also developed a coating of ZnO nanoflakes on a Pb sheet using the electrodeposition technique. They used the DC method to study the effect of two current densities, 3 mA/cm^2^ and 40 mA/cm^2^, for preparing coatings on the behaviour against *E. coli*. After being in contact surfaces and bacterial suspensions, the results indicated that both ZnO coatings exhibited strong antibacterial ability, with a greater ability for the ZnO coating generated at a higher current density, possibly due to the morphology of the ZnO crystals, which presented a smaller grain size and a greater effective contact surface between the ZnO crystals and the bacteria. Hammad et al. [211] investigated the antimicrobial activity of an electrodeposited ZnO nanocoating on NiTi alloy against two Gram-positive bacteria, *S. aureus* and *Streptococcus pyogenes* (*S. pyogenes*), and a Gram-negative bacterium, *E. coli*. Despite the fact that the NiTi alloy did not show any inhibition against bacteria, NiTi/ZnO did. Specifically, the area of inhibition in agar plates after 24 h of contact were 4.25 ± 0.49 mm, 6.25 ± 0.64 mm and 3.57 ± 0.43 mm for *S. aureus*, *S. pyogenes* and *E. coli*, respectively. As previously described by other authors, the greatest antibacterial effect took place for Gram-positive bacteria. Hui et al. [212] investigated the antibacterial capacity of a nanostructured ZnO coating with a particle size of ~25 nm. The coating was obtained by electrodeposition with sodium alginate as an additive on pure Zn. Antimicrobial activity was assessed against *S. aureus* and *E. coli* strains using the agar diffusion technique. The ZnO nanostructured coating proved to have an antibacterial capacity against both strains. It exhibited an average zone of inhibition of 24.2 mm and 25.6 mm for *E. coli* and *S. aureus*, respectively. The authors concluded that nanostructured ZnO in contact with bacteria causes Zn^2+^ to be released slowly, giving place to this inhibitory effect on bacteria.

A similar behaviour was described for ZnO covers obtained by other techniques. Rago et al. [213] manufactured ZnO microparticles with a size between 200–500 nm and nanoparticles with 20–40 nm using the hydrothermal technique, and evaluated its antimicrobial behaviour against two Gram-positive strains: *Bacillus subtilis* and *S. aureus*. The results showed that although both morphologies damaged the bacteria, the nanostructured particles were more efficient. The authors justified this result because the smaller the ZnO particle size, the greater the surface area of the bacterial membrane exposed. Particles could accumulate on the outer surface of plasma membranes neutralizing the membrane potential. This would cause an increase in the surface tension of the membrane, generating its depolarization [41]. The permeability of the cell wall would then occur, facilitating the internalization of the ZnO nanoparticles and the leakage of the intercellular fluid. Yamamoto et al. [214] have also reported the benefits of using smaller particle sizes, in the range of the nanoscale, to increase antimicrobial activity. Huang et al. [215] fabricated ZnO nanoparticles with a diameter of 60 nm, whose antimicrobial activity was evaluated against *Streptococcus agalactiae* and *S. aureus*. The results indicated that toxicity of the nanoparticles suspensions was dependent on the nanoparticle concentration: a concentration of 10^−^^1^ M caused more than a 95% inhibition of bacterial growth, while inhibition decreased down to 30% if the concentrations decreased to 1.2 × 10^−^^3^ M and 6 × 10^−^^4^ M. Moreover, a lower concentration of nanoparticles was not toxic to these microbial strains. Antibacterial activity against *E. coli* was achieved with ZnO nanoparticles between 20–60 nm in size manufactured by electrophoretic deposition by Cordero-Arias et al. [216] and likewise by Zhang et al. [40].

ZnO material is an excellent candidate for UV applications and in the visible range because of its wide bandgap of 3.37 eV [20]. It has become one of the most important semiconductors, with potential applications as a photocatalytic degradation agent and as a bacteriostatic agent under visible light [217]. Its morphology and grain size will significantly influence its activity. For example, a morphology of ZnO nanowires exhibits a bandgap of 3.27 eV with a broad emission band in the yellow–orange region [218]. Instead, a cauliflower morphology shows an emission spectrum with three photoluminescence peaks at 3.23, 3.00 and 2.30 eV [219]. Moreover, annealing treatment on a nanowire morphology modifies its forbidden band [220]. On the other hand, when the particle size or grain size of ZnO is in the order of the nanoscale, the photocatalytic and antibacterial activity will be improved [221]. This is because larger effective surface areas will ensure a wide range of photocatalytic reactions compared to smaller areas. Regarding the photocatalytic activity, Wanotayan et al. [222] evaluated the activity for a ZnO coating obtained by electrodeposition on Cu by the DC technique, with a morphology of small granules. The assessment was carried out by degradation of the methylene blue dye under a mercury lamp. This coating exhibited photocatalytic activity with a degradation efficiency of almost 100%. Lu et al. [223] also evaluated the photocatalytic activity of ZnO and ZnO/graphene coatings on a glass substrate, by the photodegradation of the methylene blue dye. The ZnO coating degraded the 32% methylene blue dye in 120 min. However, the ZnO/graphene coating showed 99.1% degradation for the same time period. The authors concluded that photocatalytic activity improved when graphene was included because of the significant change in the morphology of the coating. The ZnO crystals changed from hexagonal bars to nanorods with a size of 93.3 nm. Pauporté et al. [224] also investigated the photocatalytic activity of an electrodeposited coating of ZnO on a Zn substrate using methylene blue and congo red as dyes. In this case, the authors identified higher photocatalytic activity for thin coatings (2–3 µm) than for dense films.

When the energy absorbed by the ZnO is higher than that corresponding to its band gap, the Zn vacancy near the valence band (BV) will capture the hole, while the oxygen vacancy near the conduction band (BC) will capture the electron. The photogenerated electrons will jump from BV to BC, forming positively charged holes and negatively charged electrons on the ZnO surface. These negatively charged electrons will be able to interact with the oxygen on the ZnO surface, generating ROS such as superoxides and elemental oxygen that can give rise to the formation of superoxide radicals ($\cdot O_{2}^{-}$). In turn, the reaction of the BV hole with water molecules may also cause the formation of hydroxyl radicals (·OH) that are extremely oxidizing [217]. The most accepted antibacterial mechanism of ZnO under photocatalytic action is the production in ROS and, as previously mentioned, the release of Zn^2+^. However, Kalyani et al. [225] reported that the active oxides produced from the ZnO surface, such as H_2_O_2_ and $\cdot O_{2}^{-}$, were the main source of the antibacterial effect. Tamurejo et al. [226] fabricated a ZnO coating on an AZ31 alloy by electrodeposition. Coating was exposed to UVC irradiation with an intensity of 4.2 mW/cm^2^ for 24 h. *S. aureus* viability and adhesion to the coating before or after exposition to UV-C were tested, allowing contact between the bacteria suspension and coating for different periods of time. The quantification of the adhesion and viability of the bacteria was carried out at different contact times. The density of adhered bacteria on the ZnO-coated surface, without exposure to UV light, was around 30% and 42% lower than on AZ31 after 120 min and 300 min of contact, respectively, similarly as on the irradiated coating. The non-irradiated coating showed some damaged bacteria within the first 60 min of contact, and after 300 min of contact, more than 50% of the adherent bacteria appeared damaged. However, irradiated coating caused the viability of the bacteria to be severely compromised. A contact time of 300 min was sufficient to damage all adhering bacteria. The authors proposed that this effect was related to the high release of Mg and Zn ions from the non-irradiated substrate. However, the authors concluded that the combination of Mg and Zn ions and the ROSs produced on the surface of the ZnO coating after exposure to UVC light were behind the potent antibacterial effect described. Hong et al. [227] investigated the antibacterial capacity of Cu_2_O and Cu_2_O-ZnO composite coatings on Ni foam by electrodeposition. In comparison to Cu_2_O coatings, Cu_2_O-ZnO composite coatings showed a more compact structure and a larger effective surface because of their smaller grain size. The electrodeposited surfaces were exposed to UV light for 24 h, and next, their antibacterial activity against *E. coli* or *S. aureus* was evaluated. The compact coating exhibited an antibacterial rate of 90.23% and 88.78% for *E. coli* and *S. aureus*, respectively, but this rate was ~60% for both strains on the Cu_2_O coating. The authors concluded that this interesting antibacterial activity was associated with the formation of surface-generated hydroxyl radicals and superoxides under UV light exposure. This photocatalytic behaviour of ZnO coatings has been studied by other authors, even with the presence of other additives in the coating. Wang et al. [228] evaluated the antimicrobial behaviour against *E. coli* of two different ZnO coatings electrodeposited on glass. The first one consisted of a ZnO coating, without any other addition, and the second one consisted of a ZnO coating on which PDMS was added. Bacterial culture was placed in contact with both surfaces under UV irradiation for 210 min. Then, under darkness, substrate-bacteria suspension contact was maintained for 24 h to promote biofilm formation. The evaluation of the photocatalytic activity through the photodegradation of methylene blue indicated that both coatings exhibited a high degradation of 99.64% and 97.4% for ZnO and ZnO/PDMS, respectively, at 180 min. The number of bacteria adhered to both coatings was significantly lower compared to an uncoated glass substrate, proving the antibacterial capacity of both coatings. The number of adherent bacteria was lower in the PDMS coating, possibly due to its superhydrophobicity. Other authors reported similar behaviours of ZnO manufactured by methods other than electrodeposition. Talebian et al. [197] developed ZnO nanoparticles with different morphologies through a solvothermal method. After being exposed to UV light, they studied their antibacterial behaviour. The authors found that coatings with a cauliflower structure morphology, the smallest grain size and the largest effective surface area exhibited better antibacterial behaviour against *E. coli* and *S. aureus* than other morphologies, with a structure of hexagonal bars or spheres that generated a less effective surface. The authors concluded that the morphology, grain size and surface area of the coating were key parameters affecting the photocatalytic activity of ZnO coatings. Interest in the shape was considered by Ann et al. [229]. They studied the antibacterial activity of two different ZnO nanoparticles (ZnO-1 and ZnO-2) against two Gram-positive strains, *S. pyogenes* and *S. aureus*, and one Gram-negative strain, *Pseudomonas aeruginosa* (*P. aeruginosa*). The morphology of ZnO-1 was of a rod structure with dimensions ranging from 51 to 60 nm, and the morphology of ZnO-2 was in the form of slabs or plates with dimensions ranging 71–80 nm. Bacteria suspensions were mixed with ZnO-1 or ZnO-2 for 24 h. The percentage of bacterial inhibition was measured by optical density. The microscopy images showed that in all cases the bacteria were covered by the ZnO particles. After 8 h of contact between the particles and the bacteria, cell reduction was observed: *S. aureus*: 69% and 52% for ZnO-1 and ZnO-2, respectively; *P. aeruginosa*: 72% and 66% for ZnO-1 and ZnO-2, respectively; and *S. pyogenes*: 84 and 85% for ZnO-1 and ZnO-2, respectively. Both particles showed an antibacterial effect on the bacteria studied. In particular, the effect was stronger on *S. pyogenes*. Microscopy images showed that both *P. aeruginosa* and *S. pyogenes* appeared with damaged or even ruptured membranes. The authors suggest that the greater effect exhibited by ZnO-1 compared to ZnO-2 was possibly due to its smaller grain size morphology and greater effective surface area, which would cause a greater amount of zinc atoms and consequently a higher level of bacterial toxicity.
materials-16-05985-t003_Table 3Table 3Summary of the literature on the influence of electrochemical parameters on the antimicrobial behaviour.Ref.SurfaceCoatingCurrentTypeStrainAssayAntimicrobial Behaviour [195]Tungsten-cobaltPure Zn DC*S. aureus*Direct contactComplete inhibition of *S. aureus* growth[196]316 L stainless steelPure ZnDC*S. aureus*Direct contactComplete inhibition of *S. aureus* growth[191]CopperZnNi/TiO_2_DC *S. aureus*Agar diffusion method28 mm diameter of inhibition*E. coli*23 mm diameter of inhibition[130]Ti6Al4VHA, Mg and ZnPC(J_p_ 1.0 mA/cm^2^)*S. aureus*Agar diffusion methodIntense antimicrobial activity againts both strain with T_on_ 1 s and T_off_ 4 s. Reduced antimicrobial activity againts both strain with T_on_ 4 s and T_off_ 1 s*E. coli*[202]Carbon steelZnNi DC*E. coli*Direct contact2.92% adhered bacteriaZnNi/chitosan0.18% adhered bacteria[190]ZincZnO DC *E. coli*Direct contact1.5% adhered bacteria and intact viability DC + capsaicin0.1% adhered bacteria and 99.96% antibacterial behaviour[210]Lead ZnO DC *E. coli*Direct contactStrong antibacterial activity. Greater for 40 mA/cm^2^ vs. 3 mA/cm^2^[211]NiTi ZnO DC*S. aureus*Agar diffusion method4.25 ± 0.49 mm diameter of inhibition*S. pyogenes*6.25 ± 0.64 mm diameter of inhibition*E. coli*3.57 ± 0.43 mm diameter of inhibition[212]ZincZnO DC+Sodium alginate*S. aureus*Agar diffusion method25.6 mm diameter of inhibition*E. coli*24.2 mm diameter of inhibition[226]AZ31ZnO UVDC *S. aureus*Direct contactWithout UV. More than 50% bacteria damagedWith UV. All bacteria damaged[227]NickelCu_2_O-ZnOUVDC*S. aureus*Direct contact88.78% antibacterial rate*E. coli*90.23% antibacterial rate[228]GlassZnOUVDC*E. coli*Direct contactFew bacteria attached to both coatingsZnO-PDMS

## 5. Cell Adhesion and Proliferation of Zn Coatings

Osseointegration is fundamental for the fixation of medical implants in the human body [230,231,232,233,234]. Once an implant is placed inside the body, it is identified by the immune system as a foreign body [235] triggering a long-term inflammatory response [236]. The role of macrophages is central in the innate immune system [237], having first contact with the biomaterial and launching the body response [236,238]. These cells accumulate at the implant–tissue interface, focusing their function on phagocytizing the implant and the products released by the material degradation process [239]. Through the secretion of inflammatory cytokines involved in immune regulation and bone regulation [240,241], macrophages recruit more macrophages and other cell types, such as fibroblasts, which arrive at the implant site and secrete a collagenous matrix to form a fibrous capsule [242], causing isolation of the implant [238,242]. Zinc performs a stimulating function of osteoblastic proliferation and promotes cell differentiation through zinc-dependent synthesis of various hormones and enzymes associated with cell division and promotion, while it inhibits bone resorption caused by osteoclasts [243,244,245,246,247,248,249,250,251,252,253,254,255,256]. Several studies have shown the good biocompatibility of zinc and its ability to promote osteogenesis, both in vitro [257,258] and in vivo [247,259,260]. All these positive answers from cells make zinc a valuable candidate as a biodegradable metallic biomaterial for bone implants [261]. Table 4 shows a summary of the literature on the influence of electrochemical parameters of Zn deposition on cell adhesion and proliferation.

More than this, various authors prepared Zn coatings with other elements or molecules. In this sense, iron, as another biocompatible and biodegradable metal present in the human body, has been considered. He et al. [113] fabricated an electrodeposited coating of ZnFe alloy on a steel sheet. The authors confirmed the good biocompatibility and cell adhesion of this alloy in in vivo experiments. Histological images after implantation confirmed the formation of a layer of dense connective tissue around implant. After 4 weeks of implantation the inflammatory cells disappeared, although some macrophages, including foreign matter, were observed, probably coming from the engulfment of degradation products. Researchers proved the main role of Zn for such good biocompatibility. As implant corrosion occurs, the alkalinization of the media was suggested as beneficial for the formation of Ca-P compounds, and then favoring integration.

Nevertheless, it must be highlighted that the good biocompatibility properties of Zn are remarkable when its concentration is low to medium. The presence of a high concentration in the media can worsen its biocompatibility. Xu et al. [262] electrodeposited a monolayer of ZnFe alloy with different Zn content. The alloy with low Zn content showed a columnar microstructure with grain growth perpendicular to the substrate. The increase up to 7.2% Zn in the alloy caused grain refinement, exhibiting excellent anticorrosive properties, as the result of a decrease in i_corr_ and the corrosion rate. However, a concentration of 11.6% of Zn caused a worsening of the behaviour against corrosion, being suggested that the excess of Zn caused its precipitation. The cytocompatibility of the alloys was evaluated with human endothelial cells at a contact time with the coating of 12 and 24 h. On the coatings with low Zn concentration, in the initial stage of cell incubation (12 h), the cells showed good adhesion with very extended morphology, and at 24 h of incubation, cell morphology was good, with some intercellular connections. However, on a coating at a concentration of 11.6% of Zn, the cell spreading capacity after 12 h of incubation was deficient, and at 24 h cells were not discernible because of the layer of corrosion products deposited. These results agree with those of other authors who have shown an unfavorable effect on the promotion of bone formation in coatings with a high Zn content [263]. He et al. [188] fabricated porous scaffolds by electrodeposition of Zn onto a Fe skeleton. An vitro test with MC3T3-E1 pre-osteoblastic cells did not show toxicity for scaffolds with a concentration below 0.338 mM. In vivo assays confirmed that result. Histological images from an implantation in a rabbit femur for 3 months showed a compact and dense tissue around the skeleton of the scaffold. In this line, Ma et al. [264] reported that a Zn^2+^ concentration between 80 µM and 120 µM was detrimental to adhesion and proliferation of human vascular smooth muscle cells. Shearier et al. [258] also reported that a lethal dose of Zn^2+^ for human vascular cells was between 70 µM and 265 µM, respectively.

As previously mentioned, Zn is capable of stimulating osteoblast differentiation, which is a basic necessity for bone formation [265]. Osteoblasts are bone cells specialized in producing a bone matrix, formed by HA crystals and composed mainly of phosphate and calcium. HA coatings can be prepared on the surface of an implant by an electrochemical process [266]. However, the biomechanical stability of HA-coated implants has been shown to be poor, although they improve osseointegration [267]. For this reason, the fabrication of HA electrodeposited coatings with the inclusion of Zn (HA-Zn) has proven to be valuable for improving bone formation around the implant as well as its corrosion behaviour, since Zn^2+^ incorporated into HA coatings have been shown to increase cell viability in vitro and stimulate bone formation in vivo [268]. Yang et al. [269] evaluated the cellular response of a coating of HA and HA-Zn electrodeposited on a porous surface of pure Ti. The HA electrodeposition process generated a hexagonal pillar morphology on the substrate. However, the inclusion of Zn in the HA matrix caused the hexagon of the crystals to become more irregular and even to disappear, causing grain refinement. They incubated MC3T3-E1 mouse preosteoblastic cells for 24 h on the surface of the coating. Cellular results successfully demonstrated cell coverage on both coatings, but HA-Zn promoted proliferation and differentiation of pre-osteoblasts more successfully compared to HA alone, pointing at the superiority of the HA-Zn coating over the HA coating alone, mainly due to the presence of Zn ions. Ding et al. [270] also evaluated the cellular response of HA and HA-Zn coatings electrodeposited on a Ti surface. The incorporation of Zn in approximately 1.33%wt. to the HA network restricted nucleation and grain growth, exhibiting a softer, denser morphology and a greater grain refinement than the porous HA coating. MC3T3-E1 viability assays took place after a 1-, 3- and 5-day contact with coatings. After 5 days of culture, the authors observed a higher concentration of cells and viability on coatings which included Zn compared to those with only HA, revealing the influential positive effect of Zn inclusion. In this line, Li et al. [271] fabricated an electrodeposited coating of HA-Zn on a WE43 magnesium alloy substrate. Cytocompatibility assays with MC3T3-E1 cells demonstrated 93% and 16% viability after 6 days of contact for uncoated HA-Zn and WE43, respectively. These results showed that the HA-Zn coating enhanced proliferation of pre-osteoblasts and greatly improved the biocompatibility of the WE43 alloy.

The enhancement of cell biocompatibility because of the presence of Zn is clear in several systems. Doping a tricalcium phosphate/HA composite ceramic with Zn in amounts between 0.6%wt. and 1.2%wt. increased cell proliferation of mouse osteoblast-like cells [272]. Moreover, very low amounts of Zn, ranging from 0.012%wt. to 0.025% wt., have been shown to have a stimulating effect on bone formation in vitro and in vivo [273,274]. Similar results were obtained by other authors on cell biocompatibility with the inclusion of zinc in HA [244,272,275,276,277,278].

Even the electrodeposition of a mixed combination of metal ions with Zn could improve biocompatibility results. Huang et al. [24] electrodeposited coatings of HA co-substituted by Zn/Cu on pure Ti. The HA-ZnCu coating showed a well-compacted morphology with a refined grain size in comparison to the HA coating that showed a porous structure; this was suggested to be due to the restricted growth of the HA crystals because of the Cu and Zn species, as described by other authors [279]. MC3T3-E1 cells were seeded on the HA or HA-ZnCu coatings to assess their cytotoxicity. The biocompatibility results showed that the Zn/Cu co-substituted surface allowed higher cell growth than in the HA coating alone. Then, the authors concluded that Zn implantation is the main reason for the increased proliferation, adhesion and spread activity of MC3T3-E1 cells. Gopi et al. [130] also developed an electrodeposited coating of Mg/Zn-HA on Ti6Al4V alloy. Human osteosarcoma MG63 osteoblasts were contacted with the coatings and their morphology was evaluated after 7 days of culture. The cells spread perfectly on the surface of the coating and the original polygonal shape was maintained. These results proved that the coating exhibited excellent biocompatibility without any toxicity.

The reinforcement of HA coatings with metallic oxide nanoparticles such as ZnO, TiO_2_, Fe_3_O_4_, CuO, etc. has also been suggested by other authors. These nanoparticles are of particular interest in the medical field [280,281]. Specifically, ZnO nanoparticles have been used in the synthesis of 3D scaffolds for tissue regeneration due to their biocompatibility, osteogenesis and promotion of cell adhesion and differentiation [282]. One of the factors that seems to be decisive is the size of the nanoparticles. As they become smaller, the effective surface area increases and can determine their cytotoxicity [283]. It has been suggested that the most suitable morphology to promote osteogenesis is a porous structure with high roughness [284], with an optimum pore size between 100 μm and 500 μm [285], a micro-roughness with Ra >100 nm or a combination of nano/micro roughness [286,287,288]. This kind of adequate surface for MG-63 osteoblastic-like cells was obtained by Mehrvarz et al. [289] with a HA electrodeposited coating embedded with ZnO nanoparticles on NiTi. This coating exhibited a network structure with high volume pores, like that of bone. Rationale for this topography comes from the role of ZnO along the process of electrodeposition, since the inclusion of ZnO nanoparticles caused the roughness of the coating to increase. It is known that ZnO nanoparticles have an isoelectric point in the range of 9–10 [150]. Thus, they can form a positively charged layer on a surface at a pH below 9. Consequently, negatively charged species such as H_2_PO_4_^-^ can be attracted to the nanoparticles and transported to the cathode surface, conditioning the process of HA nucleation and growth. On the other hand, the HA electrodeposition process is associated with the formation of bubbles of H_2_ [290]. These bubbles form more intensely if the nucleation rate becomes higher. However, since bubbles will prevent the Ca-P species from diffusing in the coverage, they will nucleate around the bubbles, thus creating considerable pores. Then, after 4 days of incubation, osteoblast cells spread over the entire surface of the HA/ZnO coating, with good viability and proliferation. The authors demonstrated that the increased roughness and porosity exhibited by this coating had beneficial impacts on cell adhesion and proliferation. However, some authors have reported that the coatings composed of HA and ZnO nanoparticles can cause aggregates of ZnO nanoparticles that could reduce the cytocompatibility of the material and cause cytotoxicity [291]. To avoid this problem, Maimaiti et al. [112] electrodeposited HA/ZnO coatings with the addition of pyrrole as an additive. In this way, they were able to improve the dispersion of the ZnO nanoparticles in the HA matrix and prevent the appearance of aggregates. They incubated cells on coatings with or without pyrrole. On the coatings with pyrrole, after 1 day of incubation, the cells began to adhere correctly. After 5 days they showed good adhesion and proliferation. After 7 days, the adhesion and proliferation were even higher and with good differentiation. In contrast, cells on the HA/ZnO coating without pyrrole exhibited cell edge floating edges and worse viability.
materials-16-05985-t004_Table 4Table 4Summary of the literature on the influence of electrochemical parameters on the cell adhesion and proliferation.Ref.SurfaceCoatingCurrentTypeCellAssayCell Adhesion and Proliferation[113]SteelZnFeDCMouse pre-osteoblastic cellsHistological imagesFormation of dense connective tissue around the implant[262]Stainless steelZnFePC(J_p_ 100 mA/cm^2^, T_on_ 5 ms and T_off_ 45 ms) Mouse pre-osteoblastic cellsHistological imagesLow Zn concentrations the cell morphology was good, with intercellular connections, but increasing Zn had an unfavourable effect on cell proliferation.[188]IronPure ZnDCMouse pre-osteoblastic cellsHistological imagesNo toxicity for concentration lower than 0.338 mM. Compact and dense tissue around the scaffold [269]TitaniumHADCMouse pre-osteoblastic cellsHistological imagesCell coverage on both coatings. However, HA-Zn promoted proliferation and differentiation more successfullyHA-Zn[270]TitaniumHA DCMouse pre-osteoblastic cellsHistological imagesHigher concentration of cells and viability on coatings which included Zn compared to those with only HA.HA-Zn[271]WE43HA-ZnDCMouse pre-osteoblastic cellsHistological images93% cell viability for HA-Zn vs. 16% for uncoated WE43[24]Titanium HA/Zn-CuDC Mouse pre-osteoblastic cellsHistological imagesHA/Zn-Cu surface allowed higher cell growth than in the HA coating alone[130]Ti6Al4VMg/ HA-ZnPC(J_p_ 1 mA/cm^2^, T_on_ 4 ms and T_off_ 4 ms)Human osteosarcoma osteoblastsMTT assayExcellent biocompatibility without any toxicity[289]NiTiHA/ZnOPC(J_p_ 6 mA/cm^2^, T_on_ 1 s and T_off_ 9 s)Human osteosarcoma osteoblastsHistological imagesGood cell viability and proliferation[112]Pure TiHA/ZnOPC(pulse potential of 1.0 V/−2.5 V)Bone mesenchymal stem cellsHistological imagesBetter cell adhesion and proliferation with the Py additive

## 6. Conclusions

Zinc is a biocompatible metal present in the human body as well as a metal widely used in coatings to prevent corrosion. These two outstanding characteristics make zinc coating worthy of consideration to improve the degradation behaviour of implants. Electrodeposition is one of the most practical and common technologies to create protective zinc coatings on metals. However, different parameters involved in the electrochemical process condition the morphology, the behaviour against corrosion and, consequently, the correct functionality in medical applications of the coating.

During the manufacturing process, a higher current density causes a higher rate of nucleation and inhibition of grain growth, with a more refined morphology and better corrosion behaviour. The use of PC against DC also leads to a more refined morphology and better corrosion behaviour. This is further improved for higher J_p_ and T_off_ values, with lower T_on_. Higher temperatures generate a more uniform and fine-grained coverage with better behaviour against corrosion. However, very high values can degrade the coverage due to a greater absorption of hydrogen on the surface, which could be controllable with the addition of H_2_O_2_. A high concentration of Zn^2+^ causes a high electrodeposition rate, generating large, deposited Zn crystals. The incorporation of nanoparticles also leads to a refinement in grain size. However, in this case, the agglomeration capacity of the nanoparticles in the Zn matrix seems to be decisive to exhibit a good behaviour against corrosion.

Zinc is a promising ally for the protection of biomaterials against the bacterial invasion of their surface. An excess of Zn^2+^ can be toxic to bacterial cells. The cations interact with nucleic acids and inactivate the enzymes of the respiratory system, causing cell death. Its antibacterial capacity has been demonstrated against Gram-positive and Gram-negative cells. In its ZnO form, it has been verified to have several mechanisms of action against microorganisms, including the release of Zn^2+^ in the physiological environment, the physical alterations produced in the plasmatic membrane and the generation of ROS. In this case, its effectiveness depends on the morphology and the grain size of the coating, as well as the binding ZnO/cells. When the particle size or grain size of ZnO is in the order of the nanoscale, the photocatalytic and antibacterial activity is improved.

Zinc also performs the stimulating function of osteoblastic proliferation and promotes cell differentiation, while it inhibits bone resorption caused by osteoclasts. Nevertheless, it must be highlighted that the good biocompatibility properties of Zn are remarkable when its concentration is low to medium. Human vascular cell toxicity was detected between 70 µM and 265 µM and in MC3T3-E1 mouse cells above 0.338 mM. The fabrication of HA electrodeposited coatings with the inclusion of pure Zn has proven to be valuable for improving bone formation as well as its corrosion behaviour, since Zn^2+^ incorporated into HA coatings have been shown to increase cell viability in vitro and stimulate bone formation in vivo. The reinforcement of HA coatings with ZnO nanoparticles has also been suggested. In this case, the most suitable morphology is a porous structure with a size between 100 μm and 500 μm and with some roughness (Ra > 100 nm) or a combination of nano/micro roughness.

## Data Availability

The data presented in this study are openly available and no new data were created.

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
