# Peer review of "Electrodeposited Zinc Coatings for Biomedical Application: Morphology, Corrosion and Biological Behaviour"

_materials, 2023, doi:10.3390/ma16175985_

Round 1
Reviewer 1 Report
The authors have presented a review of electrodeposited Zn coatings for biomedical applications, and the manuscript is generally well-written. However, the following aspects need to be taken care of for improving the quality and readability-
1. The manuscript lacks tables and diagrams which can be used to convey the information in a simplified form.
2. The continuous and pulsed supply of current should be presented as separate process conditions while discussing the parameters, and the influence of the type of current supply should be discussed.
3. The authors have not included discussion on the physical, mechanical, and tribological properties, which are the primary characteristics of any coating
4. A discussion on the types of substrates used for coating and their applications should be included.
The quality of language is acceptable.
Reviewer 2 Report
The article may be useful for people involved in the study of a specific group of implants used in medicine and basically for people dealing with zinc coatings as such.
I am not sure whether the zinc coating is supposed to increase the corrosive resistance of the implants (see - Absrtact). The materials used for the implants are either very noble materials (stainless steels, titanium and alloys) that do not require sacrificial protection by zinc.
Line 60-62 The authors describe the photocatalytic effect associated with ultraviolet light. In the case of implants, this phenomenon is probably not significant for implants usually operating inside the human body.
line492 "associated with a lower Ecorr of 1.081 V and thus a more noble behaviour of the alloy" - Shouldn't there be a "-" sign here?
line 629"In these microcells, the zinc acts as an anode and the TiO2 nanoparticles act as a cathode, which facilitates anodic polarisation, and consequently only homogeneous corrosion occurs" - How to explain the formation of a microcell between zinc and TiO2. TiO2, if not properly prepared, is rather an inert substance.
The authors in the article take up a lot of thematic threads related to the current-coated zinc coatings themselves, and which are not always related to implants because they concern, for example, mild steel. As a result, the article has become very extensive and the reader can be a little confused as to what it is really about.
Round 2
Reviewer 2 Report
After the corrections made by the authors, the article is most suitable for publication. Although in the part about titanium oxide I was not completely convinced. If the material was modified to obtain a non-stoichiometric oxide with structural defects, then maybe, but pure TiO2 as a cathode is probably a slight abuse after all.